

# Total nitrate uptake by an invasive benthic foraminifer in marine sediments

Constance Choquel[1*], Emmanuelle Geslin[1], Edouard Metzger[1], Helena L. Filipsson[2], Nils Risgaard-Petersen[3], Patrick Launeau[1], Manuel Giraud[1], Thierry Jauffrais[4,1], Bruno Jesus[5,6] and Aurélia Mouret[1]

1: UMR 6112 LPG BIAF, Univ. Angers, Univ. Nantes, CNRS, France
2: Department of Geology, Lund University, Sweden
3: Department of Geosciences, Aarhus University, Denmark
4: Ifremer, IRD, Univ. Nouvelle-Calédonie, Univ. La Réunion, CNRS, UMR 9220 ENTROPIE, New Caledonia
5: Université de Nantes, Mer Molécules Santé, EA 2160, France
6: BioISI – Biosystems & Integrative Sciences Institute, Campo Grande, University of Lisbon, Faculty of Sciences, Portugal

*Correspondence to*: Constance Choquel (constance.choquel@gmail.com or constance.choquel@univ-angers.fr )

**Abstract.** Oxygen availability impacts the marine nitrogen cycle at a range of spatial and temporal scales. Invasive organisms have shown to sustainably affect sediment geochemistry and benthic ecology. *Nonionella* sp. T1 was recently described as an invasive benthic foraminifer in the North Sea region. Here, we demonstrate the impact of this denitrifying species on the foraminifera fauna and the nitrogen cycle of the Gullmar Fjord (Sweden). The foraminifera contribution to benthic denitrification was estimated by coupling living foraminifera micro-distribution, denitrification rate measurement and sedimentary nitrate 2D distribution. *Nonionella* sp. T1 dominated the foraminifera fauna and could denitrify up to 50-100 % of nitrate porewater in oxygenated bottom waters of the fjord. Contrastingly, at the deepest hypoxic low-nitrate station, denitrifying foraminifera species were scarce and did not contribute to nitrogen removal (~ 5 %). Our study showed that benthic foraminifera can be a major contributor of nitrogen mitigation in oxic coastal ecosystems and should be included in ecological and diagenetic models aiming at understanding biogeochemical cycles coupled to nitrogen.

## 1 Introduction

Hypoxic water ($[O_2] < 2$ mg $L^{-1}$ or $< 63$ µmol $L^{-1}$) occurs frequently in bottom-waters of shallow coastal seas, due to remineralization of organic matter and water stratification (e.g. Diaz et al., 2008; Breitburg et al., 2018). Hypoxia may have large ecological effects (Levin et al., 2009; Rabalais et al., 2010; Zhang et al., 2010), such as an increase of fauna mortality (Diaz et al., 2001). However, certain microorganisms, e.g. bacteria and foraminifera, can perform denitrification by respiring nitrate (Risgaard-Petersen et al., 2006) and thereby to survive in depleted oxygen environments. The effects of decreasing dissolved oxygen availability at spatial and temporal scales will impact biogeochemical cycles such as the nitrogen cycle (Childs et al., 2002; Kemp et al., 2005; Conley et al., 2007; Diaz et al., 2008; Neubacher et al., 2013; Breitburg et al., 2018). This study focus on how one important compartment of the marine meiofaunal community - the benthic foraminifera - is coupled to the nitrogen cycle during contrasted dissolved $[O_2]$ conditions, focusing on the impact of an invasive species.



The nitrogen cycle occurring in marine sediments is dependent on the bottom-water oxygenation. In oxic bottom

water conditions (Fig. 1a), ammonium ($NH_4^+$) produced from remineralization of particulate organic nitrogen (PON) in

sediments, diffuses toward the oxic sediment-superficial layer and through the water-sediment interface. Nitrification can occur

in the oxic sediment and in the oxic water column through the conversion of $NH_4^+$ to nitrate ($NO_3^-$) (Rysgaard et al., 1995;

Thamdrup and Dalsgaard, 2008). Conversely, denitrification occurs in sediment when oxygen is scarce (below 5 µmol $L^{-1}$,

Devol et al., 2008) and organic carbon and nitrate are available. Denitrification named "canonical denitrification" ($NO_3^-$ →

$NO_2^-$ → NO → $N_2O$ → $N_2$) is an anoxic process whereby nitrate is used as the terminal electron acceptor in the oxidation of

organic matter by facultative anaerobic metabolisms when oxygen is exhausted. Denitrification participates in the loss of the

fixed Nitrogen to $N_2$ gas (Brandes et al., 2007 and references within). Another process can contribute to this loss of $N_2$ gas:

Anammox (anaerobic ammonia oxidation) (Engström et al., 2005; Brandma et al., 2011). According to Brandes et al. (2007

and references within) the "total denitrification" can be defined as the sum of the canonical denitrification plus the anammox.

Nitrification and denitrification are thus strongly coupled, and denitrification can be enhanced by adjacent sedimentary

nitrification zones or by direct $NO_3^-$ diffusion from the overlying water towards the sediment (Kemp et al., 1990; Cornwell et

al., 1999). When bottom water turns hypoxic, the nitrogen cycle occurring in the sediment is strongly affected (Fig. 1 b).

Nitrate production is reduced since nitrification cannot process under low oxygen conditions. However, deeper into reduced

sediment, nitrification can occur through secondary reactions with $NH_4^+$ oxidation by Mn and Fe oxides (Luther et al., 1997;

Mortimer et al., 2004). Denitrification is the dominant process of nitrate reduction in coastal marine sediments (Thamdrup and

Dalsgaard, 2008; Herbert, 1999). However, dissimilatory nitrate reduction to ammonium (DNRA) can also contribute to nitrate

depletion in reduced sediment leading to $NO_3^-$ converstion into $NH_4^+$ instead of nitrogen ($N_2$) (Christensen et al., 2000) and

compete denitrification.

Benthic foraminifera were the first marine eukaryotes found to perform denitrification (Risgaard-Petersen et al.,

2006), but not all foraminifera species can denitrify (Piña-Ochoa et al., 2010). Denitrifying foraminifera species are defined in

our study as species able to perform denitrification proved by denitrification rate measurements. These denitrifying species

have a facultative anaerobic metabolism and nitrate-storing foraminifera can use either environmental oxygen or nitrate to

respire (Piña-Ochoa et al., 2010). *Nonionella* cf. *stella* and *Globobulimina turgida* were identified as the first denitrifying





foraminifera species (Risgaard-Petersen et al., 2006). Currently, nineteen denitrifying species are known (Glock et al., 2019).

Foraminifera denitrification rates show a large range from $7 \pm 1$ pmol N indiv. $^{-1}$ d$^{-1}$ to $2241 \pm 1825$ pmol N indiv. $^{-1}$ d$^{-1}$ (Glock et al., 2019).

Recently, *Nonionella stella* was described as an invasive species in the North Sea region and reported in the Gullmar Fjord (Sweden) (< 5 %, Polovodova Asteman and Schönfeld, 2015). However, *Nonionella stella* sampled in the Santa Barbara Basin (California USA) differs morphologically (Charrieau et al., 2018) and genetically (Deldicq et al., 2019) from the

specimens sampled in Kattegat and Oslofjord (Norway), respectively. Deldicq et al. (2019) describe these specimens as the *Nonionella* sp. T1 morphotype, a non-indigenous and invasive species in the Oslofjord. The genus *Nonionella* is potentially capable to denitrify as demonstrated with *Nonionella* cf. *stella* by Risgaard-Petersen et al. (2006). Denitrification rates of two species from the Gullmar Fjord have been measured: *Globobulimina turgida* (Risgaard-Petersen et al., 2006) and *Globobulimina auriculata* (Woehle et al., 2018). Additionaly, *Stainforthia fusiformis* and *Bolivina pseudopunctata* are two

dominant species in the deepest part of the fjord (Gustafsson and Nordberg, 2001; Filipsson and Nordberg, 2004). These species are also potential candidates for denitrification. Indeed, the denitrification rates of *Stainforthia fusiformis* from Perú were measured by Piña-Ochoa et al. (2010) and several species of *Bolivina* from Perú, Bay of Biscay and Santa Barbara were measured by Glock et al. (2019); Piña-Ochoa et al. (2010) and Bernhard et al. (2012), respectively. On the other hand, other typical fjord species such as *Bulimina marginata*, *Cassidulina laevigata, Hyalinea balthica* are considered as non-denitrifying

species by Piña-Ochoa et al. (2010) as their intracellular nitrate reserves are almost absent. The anaerobic metabolism of some other species commonly found in the fjord such as *Leptohalysis scotti*, *Liebusella goesi, Nonionellina labradorica* and *Textularia earlandi* is not documented in previous studies.

A high abundance of denitrifying foraminifera in both oxic and anoxic marine environments play an important role in the nitrogen cycle (Risgaard-Petersen et al., 2006; Piña-Ochoa et al., 2010; Bernhard et al., 2012; Glock et al., 2013; Xu et

al., 2017). Previous estimates of foraminifera contributions to denitrification range from 1 to 90 % (Dale et al., 2016; Xu et al., 2017). Estimates of foraminifera contribution to benthic denitrification are limited by the high spatial and temporal variability of sediment geochemistry and distribution of denitrifying foraminifera, which poses particular methodological challenges. Marine sediments often include chemical micro-heterogeneities (Aller et al., 1998; Stockdale et al., 2009), which

can be averaged within the volume of a sediment slice. Moreover, sediment core slicing or centrifugation can induce cell lysis,

which can induce a bias in porewater nitrate concentrations (Risgaard-Petersen et al., 2006). To characterize these microenvironments at submillimeter/ millimeter scales, new approaches have to be used. Recently, a 2D-DET (Diffusive Equilibrium in Thin-film) technique combining colorimetry and hyperspectral imagery was developed to obtain the distribution of nitrite and nitrate in sediment porewater at millimeter resolution in two dimensions (Metzger et al., 2016). This method avoids mixing of intracellular nitrate and nitrate contained in the sediment porewater.

The present study aims to examine how the invasive *Nonionella* sp. T1 and the other denitrifying species affect the nitrogen cycle by comparing two stations with contrasting oxygen and nitrate environments subjected to hypoxic events. The objectives of the paper are: (1) to characterize the density of the living benthic foraminifera at two contrasted stations; (2) to measure the denitrification rate of the invasive *Nonionella* sp. T1 and (3) to quantify its contributions to benthic denitrification; (4) to discuss the probable future impact of the invasive *Nonionella* sp. T1 on the foraminifera fauna and the nitrogen cycle in

the Gullmar Fjord.

## 2 Material and Methods

### 2.1 Site description and sampling conditions

The Gullmar Fjord is 28 km long, 1-2 km wide and located on the Swedish West coast (Fig. 2). The fjord undergoes

fluctuations between cold and temperate climates (Svansson, 1975; Nordberg, 1991; Polovodova Asteman and Nordberg, 2013; Polovodova Asteman et al., 2018). The fjord is stratified (Fig. 2 d) in four water masses (Svansson, 1984; Arneborg, 2004). Hypoxia events in the fjord have been linked to the influence of the North Atlantic Oscillation (NAO) (Nordberg et al., 2000; Björk and Nordberg, 2003; Filipsson and Nordberg, 2004). Several monitoring stations are located in the fjord: Släggö (65 m depth), Björkholmen (70 m depth) and Alsbäck (117 m depth), the hydrographic and nutrient data were obtained from

the SMHI's publically available data-base SHARK (Svenskt Havsarkiv, www.smhi.se). Since 2010, the threshold of hypoxia ($[O_2] < 2$ mg $L^{-1}$, i.e. 63 µmol $L^{-1}$) in Alsbäck station (red squares, Fig. 3) is reached typically in late autumn and winter. Deep-water exchanges usually occur in late water-early spring. However, the duration of hypoxia varies between years and hypoxia
events also occurred in the summer 2014 and 2015, due to lack of deep-water exchange. The frequency of hypoxic events has increased in the fjord (see previous studies).

Two sampling cruises were conducted in the Gullmar Fjord on board R/V *Skagerak* and *Oscar von Sydow*, respectively. The first cruise (GF17) took place between 14th and 15th November 2017 and two stations were sampled (GF17-3 and GF17-1, Fig. 2 c and d) to define the living foraminifera fauna and the sediment geochemistry at two contrasted stations. The second cruise (GF18) took place on the 5th September 2018 with the focus to collect living *Nonionella* sp. T1 for $O_2$ respiration and denitrification rates measurements. Only one station (at the same position as GF17-3) was sampled.

GF17-3 (50 m water depth) is located closest to the mouth of the fjord (58°16'50.94"N/ 11°30'30.96"E) with bottom waters from Skagerrak (blue diamond, Fig. 3) and GF17-1 (117 m depth) close to the deepest part of the fjord (58°19'41.40"N/11°33'8.40"E) near Alsbäck monitoring station in the middle of the stagnant basin (red square, Fig. 3). In November 2017, CTD profiles indicated the water mass structures at both stations (Fig. S1). Bottom water at GF17-3 station was oxic with a dissolved oxygen content of 234 µmol $L^{-1}$. The dissolved oxygen content decreased strongly with depth at the
GF17-1 station reaching 9 µmol $L^{-1}$ at the seafloor, which is below the severe hypoxia threshold.

## 2.2 Foraminifera sampling and processing

During the first cruise, two sediment cores per station (1A, 1C and 3A, 3C for GF17-1 and GF17-3 stations respectively) were immediately subsampled with a smaller cylindrical core (Ø 8.2 cm) and sliced every 2 mm up to 2 cm and every 5 mm
from 2 to 5 cm to study living foraminifera distribution. The samples were incubated without light for 10–19 hours in ambient seawater with Cell Tracker Green (CMFDA, 1 mM final concentration) at *in situ* temperatures (Bernhard et al., 2006) and then fixed with ethanol 96°. Fixed samples were sieved and the > 100 µm fraction was examined using an epifluorescence microscope equipped for fluorescein detection (i.e., 470 nm excitation; Olympus SZX13). In the present study, the foraminifera distribution will be described highlighting the invasive species *Nonionella* sp. T1.


## 2.3 Geochemical sampling and processing



One core from the shallow GF17-3 station was reserved for $O_2$ microelectrode profiling. Oxygen concentration was measured in the dark with a Clark electrode (50 µm tip diameter, Unisense ®, Denmark) within the first 5 mm depth at a 100 µm vertical resolution. Due to technical problems, no oxygen profiling was done at the GF17-1 station.

One core per station was dedicated for geochemical analyses, they were carefully brought to Lund University (Sweden) and stored at the sampling site temperature (10°C) until further analysis the next day. Overlaying water of the GF17-3 core was gently air bubbled to maintain the oxygenated conditions recorded at this station. Overlaying water of the GF17-1 core was bubbled with $N_2$ gas passed through a solution of carbonate/bicarbonate to avoid pH rise due to degassing of $CO_2$ by $N_2$ bubbling.

A summary of the $NO_2^-$/ $NO_3^-$ 2D gel method is presented in Figure 4 (details see, Metzger et al., 2016). For each core, a DET (Diffusive Equilibrium in Thin films) gel probe (16 cm x 6.5 cm and 0.1 cm thickness, Fig. 4 a) was hand-made prepared (Metzger et al., 2016). The gel probe was inserted into the sediment and left for 5 hours to allow for a diffusive equilibration time between the gel and porewaters (Fig. 4 b). After equilibration, the equilibrated gel was removed of the core and was laid on a first $NO_2^-$ reagent gel (Fig. 4 c). A reflectance analysis photograph of the nitrite gels fauna was taken with a hyperspectral camera (HySpex VNIR 1600). The next step was to convert existing nitrate into nitrite with the addition of a reagent gel of vanadium chloride ($VCl_3$) (Fig. 4 d).  After 20 min at 50°C, a pinkish coloration appeared revealing porewater nitrate concentration (Fig. 4 e). Followed by the acquisition of another hyperspectral image and converted into false colours through a calibrated scale of concentrations, the final image was cropped to avoid border effects (Fig. 4 f). Each pixel (190 µm x 190 µm) was decomposed as a linear combination of the logarithm of the different end-member spectra using ENVI software (unmixing function) (Cesbron et al., 2014; Metzger et al., 2016). Nitrite and nitrate detection limit is 1.7 µmol $L^{-1}$ (Metzger et al., 2016). Nitrate production/consumption zones for each station were estimated by extracting the average and standard deviation of the 290 vertical 1D profiles ((5.5 cm width x 1 pixel) / 0.019 cm for 1-pixel size) on the 2D gels and modelling using PROFILE software (Berg et al., 1998).

## 2.4  Oxygen respiration and denitrification rates measurements of the invasive *Nonionella* sp. T1

The two cores sampled in the 2nd cruise (GF18, September 2018) at the shallower GF17-3 station were carefully transported at *in situ* temperature (8 °C) and stored for three days at the Department of Geosciences, Aarhus University (Denmark). *Nonionella* sp. T1 specimens were picked under *in situ* temperature and collected in a Petri dish, containing a thin layer of sediment (32 µm) to check their vitality. Only living, active *Nonionella* sp. T1 specimens were picked and cleaned

several times using a brush with micro-filtered, nitrate-free artificial seawater.

Oxygen respiration rates were measured, following the method developed by Høgslund et al. (2008) using a Clark type oxygen microsensors (50 µm tip diameter, Unisense ®, Denmark) (Revsbech, 1989) calibrated by a two-point calibration using air-saturated water at *in situ* temperature (8 °C) and sodium ascorbate solution (to strip $O_2$ out of the system) as zero. Then, a pool of 5 living *Nonionella* sp. T1 was transferred into a glass microtube (inner diameter 0.5 mm, height 7.5 mm) that

was fixed inside a 20 ml test tube mounted in a glass-cooling bath (8 °C). A motorized micromanipulator was used to measure $O_2$ concentration profiles along a distance gradient that ranged from 200 µm of the foraminifera to 1200 µm using 100 µm steps. Seven $O_2$ concentration profiles were generated with one incubation containing the pool of *Nonionella* sp. T1. Negative controls were done by measuring $O_2$ rates from microtube with empty foraminifera shells and blanks with empty microtube. Oxygen respiration rates were calculated with Fick's first law of diffusion, $J = -D * dC/dx$, where J is the flux, dC/dx is the

concentration gradient obtained by profiles and D is the free diffusion coefficient of oxygen at 8 °C for a salinity of 34 (1.382 x $10^{-5}$ $cm^{-2}$ $s^{-1}$, Ramsing and Gundersen, 1994). The seven $O_2$ respiration rates were calculated as the product of the flux by the cross section area of the microtube (0.196 $mm^2$). Then, the average $O_2$ respiration rate was divided by the 5 *Nonionella* sp. T1 presented in the microtube to obtain the respiration rate per individual.

The same pool of *Nonionella* sp. T1 specimens as for the $O_2$ respiration rates was used for denitrification rate measurements. Denitrification rates were measured as it is described in Risgaard-Petersen et al., (2006). In this method, denitrification is stopped at the $N_2O$ production by acetylene inhibition that can be measured with a $N_2O$ microprobe (50 µm tip diameter, Unisense ®, Denmark). Thus, $N_2O$ was measured as the end product instead of $N_2$ (Risgaard-Petersen et al., 2006).

Nitrous oxide flux was estimated from the chemical gradient profiled from the pool of *Nonionella* sp. T1 inserted in

a microchamber. The $N_2O$ production was multiplied by two because two moles of $NO_3^-$ are required for the production of one

mole of $N_2O$ (Risgaard-Petersen et al., 2006). The microchamber is porous to gases and is bathed in a sodium ascorbate solution

that maintains oxygen concentration at zero within the microchamber. The microchamber was filled with an oxygen/nitrate-

free solution of artificial seawater saturated with acetylene (to inhibit $N_2O$ transformation into $N_2$) containing 5 mM of Hepes

buffer (to maintain the pH stable). Calibration was performed using the standard addition method by successive injections of

a $N_2O$ saturated solution in order to have 14 μM steps of final concentration. Negative controls were done by checking the

absence of $O_2$ from microchamber with empty foraminifera shells and blanks with empty microchamber. Then, the pool of

*Nonionella* sp. T1., was transferred to the microchamber with a micropipette. The $N_2O$ concentration profiles were repeated

seven times on the pool of *Nonionella* sp. T1. The source of nitrate during denitrification comes from intracellular nitrate

storage of *Nonionella* sp. T1 (not measured in this study).

       Since $O_2$ respiration and denitrification rates are linked to cytoplasmic volume or biovolume (BV) (Geslin et al., 2011;

Glock et al., 2019), the specimens from the pool of *Nonionella* sp. T1 were measured (width (a) and length (b) Fig. 5) using a

micrometer mounted on a Leica stereomicroscope (MZ 12.5) to estimate the average BV. The volume of the shells was

estimated by using the best resembling geometric shape, a spheroid prolate ($V = \frac{4}{3}\pi\left(\frac{a}{2}\right)^2\left(\frac{b}{2}\right)$). Then, according to Hannah et

al., (1994) 75 % of the measured entire volume of the shell was used corresponding to the estimated cytoplasmic volume. To

compare the size of the *Nonionella* sp. T1 sampled in the 1st cruise (GF17, study of the fauna) with the *Nonionella* sp. T1

samples in the 2nd cruise (GF18, denitrification rate measurements), 5 specimens sampled in the 1st cruise were also measured.

## 2.5   Contributions of the invasive *Nonionella* sp. T1 to diffusive oxygen and nitrate uptake

The following estimated contributions to sediment diffusive oxygen and nitrate uptake were performed mainly on the

dominant denitrifying species, *Nonionella* sp. T1. The size of the *Nonionella* sp. T1 specimens sampled during the two cruises

differed markedly (Table S1). Thus, we need to correct the denitrification rate of *Nonionella* sp. T1 specimens from the 1st

cruise to take into account the difference of shell size. Thus, the measured *Nonionella* sp. T1 denitrification rate (2nd cruise)

was normalized by specimen BV (1st cruise) using the relationship: ln (y) = 0.68 ln (x) – 5.57, where y is the denitrification

rate (pmol ind$^{-1}$ d$^{-1}$) and x is the shell BV ($\mu$m$^3$) ((Geslin et al., 2011; Glock et al., 2019; Equation S1). The corrected *Nonionella*

sp. T1 denitrification rate is multiplied by the *Nonionella* sp. T1 specimens counted found in each denitrifying zones defined

by PROFILE modelling. Then, two calculation approaches were discussed to estimate *Nonionella* sp. T1 contributions to

benthic denitrification: (A) to divide the *Nonionella* sp. T1 denitrification rate by the nitrate porewater denitrification rate

estimated from PROFILE modelling, then the second calculation (B) to divide the *Nonionella* sp. T1 denitrification rate by the

total denitrification from PROFILE plus the *Nonionella* sp. T1 denitrification rate. In the first approach (A) we suggest

*Nonionella* sp. T1 use only the nitrate in the sediment porewater. In the second approach (B) we suggest that the foraminifera

only use both intracellular and porewater nitrate pool for denitrification.

### 3    Results

**3.1   The invasive *Nonionella sp.* T1 O$_2$ respiration and denitrification rates in the Gullmar Fjord**

The O$_2$ respiration rates measured in the pool of *Nonionella* sp. T1 specimens collected in the 2$^{nd}$ cruise (GF18) were 169

$\pm$ 11 pmol O$_2$ indiv$^{-1}$ d$^{-1}$ with an average BV of 1.3 $\pm$ 0.7 10$^{+06}$ $\mu$m$^3$ (BV details, Table S1). The denitrification rate, measured

on the same pool of specimens, was 21 $\pm$ 9 pmol N indiv$^{-1}$ d$^{-1}$.

The *Nonionella* sp. T1 average BV collected in the 1$^{st}$ cruise (GF17-3) was 4.0 $\pm$ 0.6 10$^{+06}$ $\mu$m$^3$, i.e. more than three times

larger the *Nonionella* sp. T1 average BV from the 2$^{nd}$ cruise (1.3 $\pm$ 0.7 10$^{+06}$ $\mu$m$^3$). As denitrification rates and foraminifera

BV are linked (see method), the measured denitrification rate was corrected using the BV of Nonionella sp. T1 from the 1$^{st}$

cruise. Thus, the *Nonionella* sp. T1 corrected denitrification rate was 38 $\pm$ 8 pmol N indiv$^{-1}$ d$^{-1}$ (Equation S1).

**3.2   The invasive *Nonionella sp.* T1 and foraminifera fauna regarding porewater nitrate micro-distribution**

The bottom water at GF17-3 station was oxic (Fig. S1, [O$_2$] = 234 $\mu$mol L$^{-1}$) and the measured oxygen penetration depth

(OPD) in the sediment was 4.7 $\pm$ 0.2 mm (n = 3). No nitrite was revealed on the gel (< 1.7 $\mu$mol L$^{-1}$), only nitrate was detected.

Bottom water average NO$_3^-$ concentration was 14.6 $\pm$ 2.3 $\mu$mol L$^{-1}$ and nitrate concentration decreased with depth in the

sediment (Fig. 6 c, d). Nitrate concentration ranged between 13.1 $\pm$ 3.2 to 11.7 $\pm$ 3.4 $\mu$mol L$^{-1}$, from the water-sediment



interface to the OPD. Nitrate concentration decreased strongly after the OPD from $11.7 \pm 3.4$ to $2.8 \pm 0.9$ µmol L$^{-1}$ until 4.0

cm depth. From 4.0 to 5.0 cm depth NO$_3^-$ concentration was very low with an average value of $2.7 \pm 0.9$ µmol L$^{-1}$ (Fig. 6 c ,

d). The PROFILE parameters (Berg et al., 1998) used on laterally averaged nitrate porewater vertical distribution of both

stations are available in Table S2. Thus, the PROFILE modelling of the averaged nitrate porewater profiles revealed one

nitrification zone from 0 to 1.2 cm depth and two denitrifying zones (red line, Fig. 6 d). The first denitrification zone occurred

between 1.2 to 3.6 cm depth with a nitrate consumption of $3.39 \ 10^{-07}$ µmol m$^{-2}$ d$^{-1}$ and the second smaller consumption zone

was from 3.6 to 5 cm depth ($1.32 \ 10^{-08}$ µmol m$^{-2}$ d$^{-1}$). The total denitrification rate from 1.2 to 5 cm depth was $3.52 \ 10^{-07}$ µmol

m$^{-2}$ d$^{-1}$ (Fig. 6 d).

The total densities of living foraminifera were similar between the cores GF17-3A and 3C (Ø 8.2 cm, 5 cm depth) with

1256 individuals and 1428 individuals, respectively (Fig. 6 a and b; Table S3, GF17-3A and 3C). *Nonionella* sp. T1 was the

main denitrifying species, accounting for 34 % of the total living fauna in the core GF17-3A and 74 % in GF17-3C (Fig. 6 a,

b; Table S4). One other candidate to denitrification, *Stainforthia fusiformis*, was found in the core GF17-3A and 3C in minority:

1 % of the total fauna in both cores (Fig. 6 a, b; Table S4, GF17-3A and 3C). The other known denitrifying species previously

reported in the Gullmar Fjord, *Globobulimina turgida* (Risgaard-Petersen et al., 2006) and *Globobulimina auriculata* (Whoele

et al., 2018) were absent. Three non-denitrifying species (Piña-Ochoa et al., 2010; Xu et al., 2017; Glock et al., 2019) were

dominant in the cores GF17-3A and 3C: *Bulimina marginata* (37 and 5 %, respectively)*, Cassidulina laevigata* (9 and 5 %)

and *Leptohalysis scotti* (11 and 9 %).

The density and the micro-distribution of *Nonionella* sp. T1 differed between the two cores (Fig. 6 a and b; Table S3,

GF17-3A and 3C). In the core GF17-3A and 3C respectively, *Nonionella* sp. T1 density showed large variability from the

water-sediment interface to 1.2 cm depth (Table S3, GF17-3A and 3C) where *Nonionella* sp. T1 relative abundance accounted

for 18 % and 50 % of the fauna in the nitrification zone (Table S4, GF17-3A and 3C). In the first denitrifying zone from 1.2

250 cm to 3.6 cm the *Nonionella* sp. T1 relative abundance represented 27 % and 78 % of the fauna. In the second denitrifying

zone, the *Nonionella* sp. T1 relative abundance increased from 3.6 to 5 cm depth and dominated the fauna by 60 % and 98%.

The relative abundance of the denitrifying candidate, *Stainforthia fusiformis*, was a minor component in each zones of both

cores and did not exceed 2 % (Table S4, GF17-3A and 3C). The three non-denitrifying species (e.g. *Bulimina marginata*,





*Cassidulina laevigata* and *Leptohalysis scotti*) also dominated the fauna of both cores GF17-3A and 3C (Table S3 and S4,

GF17-3A and 3C). From the water-sediment interface to 1.2 cm depth (0-1.2 cm depth) *B. marginata* accounted for 42 % and

12 %, *C. laevigata* 16 % and 13 % and *L. scotti* 6 % and 11 %, respectively. In the first denitrifying zone (1.2-3.6 cm depth)

*B. marginata* accounted for 34 % and 2%, *C. laevigata* 7 % and 2% and *L. scotti* 25 % and 13 %, respectively. In the second

denitrifying zone (3.6-5 cm depth) *B. marginata* accounted for 34 % and 0 %, *C. laevigata* was absent and *L. scotti* 5 % and

1 %, respectively.


Due to severe hypoxia at the GF17-1 station, oxygen was assumed to be below detection limit within the sediment. No

nitrite was detected at this station (< 1.7 µmol L$^{-1}$). Average NO$_3^-$ concentration in the bottom water reached 5.7 ± 1.0 µmol L$^{-1}$

(Fig. 6 g and h). Nitrate concentrations decreased from the sediment surface (4.2 ± 1.0 µmol L$^{-1}$) to 1.6 cm (1.8 ± 0.6 µmol L$^{-1}$)

and then average nitrate concentration remained below the detection limit (1.7 µmol L$^{-1}$). However, a patch with higher

nitrate concentration was visible on the left part of the gel between 2.0 and 3.0 cm depth. A 1D vertical profile passing through

this patch (white line, Fig. 6 g) was extracted from the 2D image and the maximal nitrate concentration of the patch was above

the detection limit with a value of 6.5 µmol L$^{-1}$ at 2.3 cm depth (blue squares profile, Fig. 6 h). The PROFILE modelling

(parameter details Table S2) of the laterally averaged nitrate vertical distribution revealed at the sampling time one denitrifying

zone from the surface to 1.6 cm depth with a nitrate consumption of 2.34 10$^{-07}$ µmol m$^{-2}$ d$^{-1}$ (red line, Fig. 6 h). Below 1.6 cm

depth, nitrate concentration was below the detection limit (hatched grey zone, Fig. 6 h), thus no PROFILE modelling was done

after this depth.

Living foraminifera showed different total densities and a large difference in species distribution between the two cores

GF17-1A and 1C (Fig. 6 e, f; Table S3, GF17-1A and 1C), with 1457 individuals and 786 individuals, respectively (Ø 8.2, 5

cm depth). *Nonionella* sp. T1 represented a low relative abundance of the total fauna with 5 % in the core GF17-1A and was

almost absent (1 %) in GF17-1 C (Table S4, GF17-1A and 1C). The known denitrifying *Globobulimina auriculata* was minor

in the fauna 1 % and 2%. The denitrifying candidate *Stainforthia fusiformis* was also found in the cores GF17-1A and 1C

reaching only 3% of the total fauna (Figure 6 e, f; Table S4, GF17-1A and 1C). The other denitrifying candidate *Bolivina*

*pseudopunctata*, was almost absent of the total fauna 0 % and 2 % (Table S4, GF17- 1A and 1C). The same three non-





denitrifying species as for the oxic station were also dominant in both cores GF17-1A and 1C: *Bulimina marginata* (64 and 30

%)*, Cassidulina laevigata* (16 and 15 %) and *Leptohalysis scotti* (4 and 36 %).

In the denitrifying zone (0-1.6 cm) *Nonionella* sp. T1 relative abundance was low, with 2 % in the core GF17-1A and was almost absent from the fauna in GF17-1C. In the core GF17-1A, *Nonionella* sp. T1 relative abundance reached 26 % of the fauna between 1.4 and 2.5 cm depth (Fig. 6 e, GF17-1A), whereas it was almost absent from the rest of the core GF17-1A and was absent from the core GF17-1C (Table S4). In the cores GF17-1A and 1C, *S. fusiformis* reached respectively 2 % and

3 % in the denitrifying zone (0-1.6 cm). In the rest of the cores from 1.6 to 5 cm depth, *S. fusiformis* represented 4 and 1 % of the fauna, respectively.  The three other non-denitrifying species dominated both cores GF17-1A and 1C. In the denitrifying zone (0-1.6 cm depth) *B. marginata* accounted for 66 % and 35 %, *C. laevigata* 19 % and 19 % and *L. scotti* 4 % and 24 %. From 1.6 to 5 cm depth, *B. marginata* dominated the fauna by 61 % and 11 %, *C. laevigata* 5 % and 2 % and *L. scotti* 6 % and 75 %, respectively.


## 4  Discussion

### 4.1  Towards a change in living foraminifera fauna of the Gullmar Fjord?

The presence and relative abundance of *Nonionella* sp. T1 in the Gullmar Fjord and in the Skagerrak-Kattegat strait has been documented during the last decades. The earliest SEM observations of specimens resembling *Nonionella* sp. T1

morphotype in the deepest part of the fjord date back to summer 1993 (identified as *Nonionella turgida*, Gustafsson and Nordberg, 2001). The invasive characteristic of *Nonionella stella* was firstly demonstrated by Polovodova Asteman and Schönfeld, (2015). Then, *Nonionella stella* was named *Nonionella* sp. T1 morphotype also described as invasive by Deldicq et al. (2019). The estimated introduction date of the invasive species into the deepest part of the fjord is 1985 according to Polovodova Asteman and Schönfeld, (2015). The relative abundance of the invasive species in the deepest fjord station was

less than 5 % between 1985 and 2007 (Polovodova Asteman and Schönfeld, 2015 and references within). At the GF17-1 hypoxic station, the *Nonionella* sp. T1 relative abundance was between 1-5 % (Table S4, GF17-3A and 3C). Thus, the *Nonionella* sp. T1 relative abundance in the deepest part of the fjord seems to remain stable. Whereas, at the GF17-3 oxic station, closest to the mouth of the fjord, the relative abundance of *Nonionella* sp. T1 varied between 34 and 74 % (Table S4,



GF17-3A and 3C). Previous studies showed an increase in the relative abundance of *Nonionella* sp. T1 morphotype in the

Skagerrak-Kattegat region (near the entrance of the Gullmar Fjord). The invasive species represented 10 % of the fauna in

June 2013 (Polovodova Asteman and Schönfeld, 2015) and up to 26 % in November 2013 (Charrieau et al., 2018). The

foraminifera fauna in the Gullmar Fjord has changed over the last decennium and *Nonionella* sp. T1 has become an important

part of the foraminifera fauna in the fjord oxic zones.

The foraminifera fauna found in November 2017 in the fjord (our results) differed from previous studies. Indeed, until the

early 1980s, the foraminifera fauna in the deepest part of the fjord was dominated by a typical Skagerrak – Kattegat fauna

(*Bulimina marginata, Cassidulina laevigata, Hyalinea balthica, Liebusella goësi, Nonionellina labradorica* and *Textularia*

*earlandi*) (Nordberg et al., 2000). However, the fauna changed. *Stainforthia fusiformis* and *Bolivina pseudopunctata* became

the major species (Nordberg et al., 2000; Filipsson and Nordberg, 2004). Further studies by Polovodova Asteman and

Nordberg, (2013) demonstrated that at least until 2011 *S. fusiformis*, *B. pseudopunctata* and *T. earlandi* dominated the fauna.

Foraminifera fauna described in the present study differ are the consequences of the occurrence of numerous severe hypoxic

events in the fjord (Fig. 3) due to lack of deep-water exchange. In November 2017 *S. fusiformis* did not exceed 3 % of the

fauna (Table S4, GF17-1A and 1C), *B. pseudopunctata* reached only 2 % in the core GF17-1C (Table S4, GF17-1C) and *T.*

*earlandi* was a minor species < 1 %.  Then, in November 2017 *Bulimina marginata, Cassidulina laevigata* and *Leptohalysis*

*scotti* were the dominant species in the fjord, ranging between 5-64 %, 5-16 % and 4-37 % of the total fauna. The *Elphidium*

*clavatum-selseyensis* species complex (following the definition from Charrieau et al., 2018), *Hyalinea baltica*, *Nonionellina*

*labradorica,* and *Textularia earlandi* were present in low relative abundance (< 5 %, Table S4). Namely, *Globobulimina*

*turgida* reached 37 % of the foraminifera fauna in August 2005 at the deepest station (Risgaard-Petersen et al., 2006); whereas

in November 2017 this species was minor. The decreasing in relative abundance of *Stainforthia fusiformis* and *Bolivina*

*pseudopunctata* must be interpreted with caution since our study used the > 100 µm fraction whereas some of the previous

studies used > 63 µm. We also wet picked the specimens and used Cell Tracker Green to identify living foraminifera, which

might affect the results compared to Rose Bengal studies of dry sediment residuals. The relative abundance of the invasive

*Nonionella* sp. T1 has increased since the study of Polovodova Asteman and Schönfeld, (2015) in the oxic part of the fjord.

The two non-denitrifying species *Bulimina marginata* and *Cassidulina laevigata* described as typical species of the Skagerrak-





Kattegat fauna (Filipsson and Nordberg, 2004) have again increased markedly in the fjord. It is evident that the foraminifera

fauna in the Gullmar Fjord is presently very dynamic with considerable species composition shifts.

## 4.2 The invasive *Nonionella* sp. T1 ecology considering the nitrate micro-distribution at the oxic station

Our study showed, for the first time, *Nonionella* sp. T1 dominated the foraminifera fauna in the Gullmar Fjord, this at the

GF17-3 oxic station despite some spatial variability (Fig. 6 a, b; Table S3; S4, GF17-3). *Nonionella* sp. T1 density increased

with sediment depth below the oxic zone (Fig. 6 a – d; Table S3, GF17-3), which could be explained by its preference to

respire nitrate rather than oxygen. This would be following the hypothesis of using nitrate as a preferred electron acceptor

suggested by Glock et al., (2019). *Nonionella* sp. T1 distributions could be explained by its capacity to store nitrate

intracellularly before porewater nitrate was denitrified by other organisms such as bacteria. At this station, *Nonionella* sp. T1

distributions may be explained as: following the oxic zone (Fig. 6 c, d; from the surface to OPD) *Nonionella* sp. T1 respires

oxygen ($169 \pm 11$ pmol $O_2$ indiv$^{-1}$ d$^{-1}$). Deeper in the hypoxic zone containing nitrate (Fig. 6 c, d; from OPD to 3.6 cm depth),

*Nonionella* sp. T1 accumulates intracellular nitrate and respires nitrate ($38 \pm 8$ pmol N indiv$^{-1}$ d$^{-1}$). In the hypoxic zone where

the nitrate porewater is depleted (Fig. 6 c, d; from 3.6 to 5 cm depth) *Nonionella* sp. T1 respires on its intracellular nitrate

reserves to survive (Fig. 6 a, b; from 3.5 to 5 cm depth). When the intracellular nitrate reserve runs out, *Nonionella* sp. T1 can

migrate to an upper zone where nitrate is still present in the sediment to regenerate its intracellular nitrate reserve (Fig. 6 a, b;

from 1.2 to 3.5 cm depth).

## 4.3 The foraminifera ecology considering the nitrate micro-distribution at the hypoxic station

Hypoxia occurred approximately at least one month before the sampling cruise in the deepest part of the fjord (Fig. 3).

When hypoxia is extended to the water column, nitrification both in the water column and the sediments is reduced or even

stopped, as oxygen is almost absent (Fig. 1 b; Childs et al., 2002; Kemp et al., 2005; Conley et al., 2007; Jäntti and Hietanen,

2012). Under this condition, the coupled nitrification-denitrification processes are strongly reduced (Kemp et al., 1990). At the

GF17-1 station, no nitrification in superficial sediment was showed by our data and nitrate was low but still detectable in the



bottom water. Nitrate can diffuse from the water column into the sediment, and thereby generate the denitrification zone as modelled by PROFILE between the surface and 1.6 cm depth (Fig. 6 h).

The rare presence of the invasive *Nonionella* sp. T1 and other denitrifying species as *Globobulimina auriculata, Bolivina pseudopunctata* and *Stainforthia fusiformis* in the hypoxic station indicate that sediment chemical conditions turned unfavorable towards denitrification during prolonged hypoxia. Instead, the non-denitrifying species *Bulimina marginata, Cassidulina laevigata,* and *Leptohalysis scotti* dominated in this hypoxic environment. Their survival could be due to seasonal dormancy (Ross and Hallock, 2016; LeKieffre et al., 2017). The suspected deep nitrification zone (blue square profile, Fig. 6

h) could explain the presence of nitrate micro-niches deeper in the sediment and might explain the patchy distribution of *Nonionella* sp. T1 also at the hypoxic site (see Fig. 6 e; Table S3, GF17-1A). Therefore, deep nitrate production in these micro-environments could favor the presence of *Nonionella* sp. T1, which can be attracted by this nitrate source as a electron acceptor to respire (Nomaki et al., 2015; Koho et al., 2011). This deep nitrification zone could be a result of an aerobic or anaerobic process. An aerobic nitrification zone in deep sediment can be formed by macrofaunal activity (burrowing activity) that

introduce some oxygen deeper into anoxic sediment (Aller, 1982; Karlson et al., 2007; Nizzoli et al., 2007; Stief, 2013; Maire et al., 2016). This nitrification zone could also be due to an anaerobic process. The Gullmar Fjord is Mn-rich (Goldberg et al., 2012) and metal-rich particles can be bio-transported into the anoxic sediment, thus allowing ammonium oxidation into $NO_3^-$ by Mn and Fe-oxides in the absence of oxygen deeper in the sediment (Aller, 1994; Luther et al., 1997).

**4.4   Contributions and potential impacts of the invasive *Nonionella* sp. T1 to benthic denitrification in the Gullmar Fjord**

If we consider that *Nonionella* sp. T1 is denitrifying the nitrate from sediment porewater (approach A, Table 1; see method 2.5) its contribution to benthic denitrification in the oxic station would be 46 % in the core GF17-3A and would reach 100 % in the core GF17-3C. If we consider that *Nonionella* sp. T1 also uses its intracellular nitrate pool for denitrification

(approach B), its contribution to benthic denitrification would be 32 % in the core GF17-3A and would reach 50 % in the core GF17-3C (Table 1). These two calculation approaches highlight the difficulties and the importance of knowing the concentration of environmental nitrate and foraminifera intracellular nitrate at the same time to estimate at best the





contributions of foraminifera to benthic denitrification. Moreover, in this study there is no data on anammox process which contributes also in the total denitrification (Brandes et al., 2007). The results reported in previous studies as Engström et al.,

(2005) do not allow us to extrapolate their data at our oxic station, located at the entrance to the fjord. Thus, we assume that our estimate of denitrification is conservative, since the possible contribution of anammox is not included in the calculation. However, despite these uncertainties *Nonionella* sp. T1 contributions to benthic denitrification support the hypothesis that this invasive denitrifying foraminifer play a major role in the benthic nitrogen cycle for sediments showing nitrification processes. At the hypoxic station, the opposite was shown where the estimated contribution of *Nonionella* sp. T1 to benthic denitrification

was below 1 % whatever the calculation approach. The estimated contributions of the other denitrifying foraminifera found in the hypoxic station were low. *Stainforthia fusiformis* did not exceed 5 %, *Globobulima auriculata* and *Bolivina pseudopunctata* were scarce and their contributions to benthic denitrification were negligible. Foraminifera contributed to almost 5 % of benthic denitrification in the hypoxic station. Compared to the oxic station, the invasive *Nonionella* sp. T1 and the other denitrifying species contributions to benthic denitrification were small in a prolonged hypoxic station of the Gullmar Fjord.


Overall, the Gullmar Fjord is well oxygenated except for the deepest basin where oxygen goes down when there is no deep water exchange (Fig. 3 c). Therefore, the GF17-3 oxic station could be considered more representative of the Gullmar Fjord benthic ecosystem. *Nonionella* sp. T1 is not the most efficient denitrifying species compared to *Globobulimina turgida* (42 pmol N ind$^{-1}$ d$^{-1}$, with BV = 1.3 10$^{+06}$ µm$^3$) and also less efficient than *Nonionella* cf. *stella* from Perú. However, *Nonionella*

sp. T1 high density could accelerate sediment denitrification and participate to increase the contrast between the two hydrographic conditions. Indeed, an increase in contrast due to oxygenation conditions: oxic vs severe hypoxia induced a gap in the availability of nitrate for anaerobic facultative metabolisms in the sediment. In the oxygenated part of the fjord, high contribution to benthic denitrification (estimated between 50 and 100%) by *Nonionella* sp. T1 could contribute to the de-eutrophication of the system by increasing the N$_2$ loss. Thus, the high densities of denitrifying foraminifera as *Nonionella* sp.

T1 would be rather beneficial. Whereas, in the hypoxic parts of the fjord, nitrate and nitrite rapidly exhausted become scarce, resulting in a decrease in denitrification. The consequence is a decrease of denitrifying foraminifera fauna. The increase of ammonium in anoxic sediment resulting by a decrease in nitrification, denitrification and anammox processes does not allow

Biogeosciences Open Access
Discussions
EGU

the nitrogen elimination from the sediment to the water column, thus potentially promoting eutrophication of the fjord in parts subjected to prolonged severe hypoxia (Fig. 1). Moreover, the low availability of nitrate in the sediment would possibly

increase the benthic transfer towards the water column of reduced compounds such as manganese and iron produced deeper in the sedimentary column by other anaerobic metabolisms (Hulth et al., 1999). These new results demonstrate that the role of denitrifying foraminifera is underestimated in the nitrogen cycle and overlooking this part of the meiofauna may lead to a misunderstanding of environments subject to hydrologic changes.

**5   Conclusion**

This study revealed a drastic change in living foraminifera fauna due to several hypoxic events that occurred in the last decennium in the Gullmar Fjord. For the first time, the invasive *Nonionella* sp. T1 dominated up to 74 % the foraminifera fauna at a station with oxygenated bottom waters. This invasive species can denitrify up to 50-100 % of the nitrate porewater sediment under oxic conditions in the fjord. Whereas, under prolonged hypoxia, nitrate depletion turns environmental

conditions unfavorable for foraminifera denitrification, resulting in a low density of *Nonionella* sp. T1 and other denitrifying species. Thus, foraminifera contribution to benthic denitrification was negligible (~ 5 %) during prolonged seasonal hypoxia in the fjord. Moreover, the invasive denitrifying *Nonionella* sp. T1 could impact the nitrogen cycle under oxic conditions by increasing the sediment denitrification and could counterbalance potential eutrophication of the fjord. Thus, our study demonstrated that the role of denitrifying foraminifera is underestimated in the nitrogen cycle especially in oxic environments.






**Figures list**

(a) Oxic bottom water

(b) Hypoxic bottom water

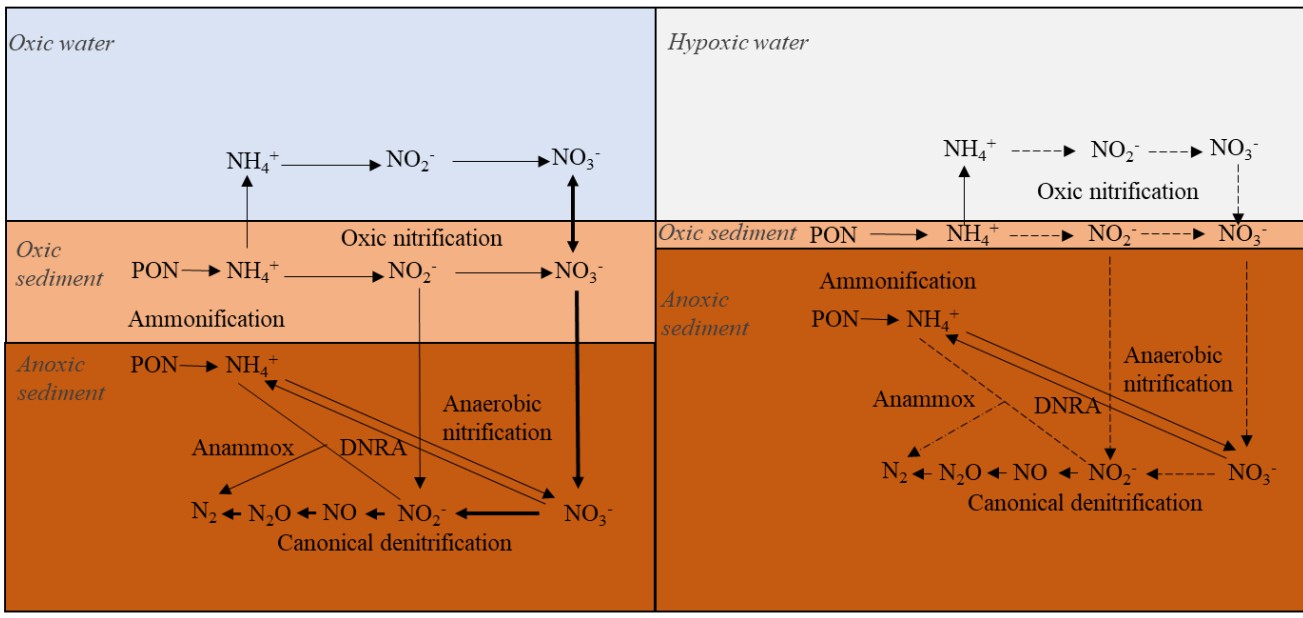

**Figure 1. Simplified nitrogen cycling in marine sediments when the bottom water is oxic (a) and hypoxic (b). Chemical formulae: PON (particulate organic nitrogen), $NH_4^+$ (ammonium), $NO_3^-$ (nitrate), $NO_2^-$ (nitrite), NO (nitrogen oxide), $N_2O$ (nitrous oxide), $N_2$ (nitrogen). The bold/dotted arrows indicate reactions advantaged/reduced by oxygen and nitrate presence/depletion. See text for more details. Modified from Jantti and Hietanen, (2012).**





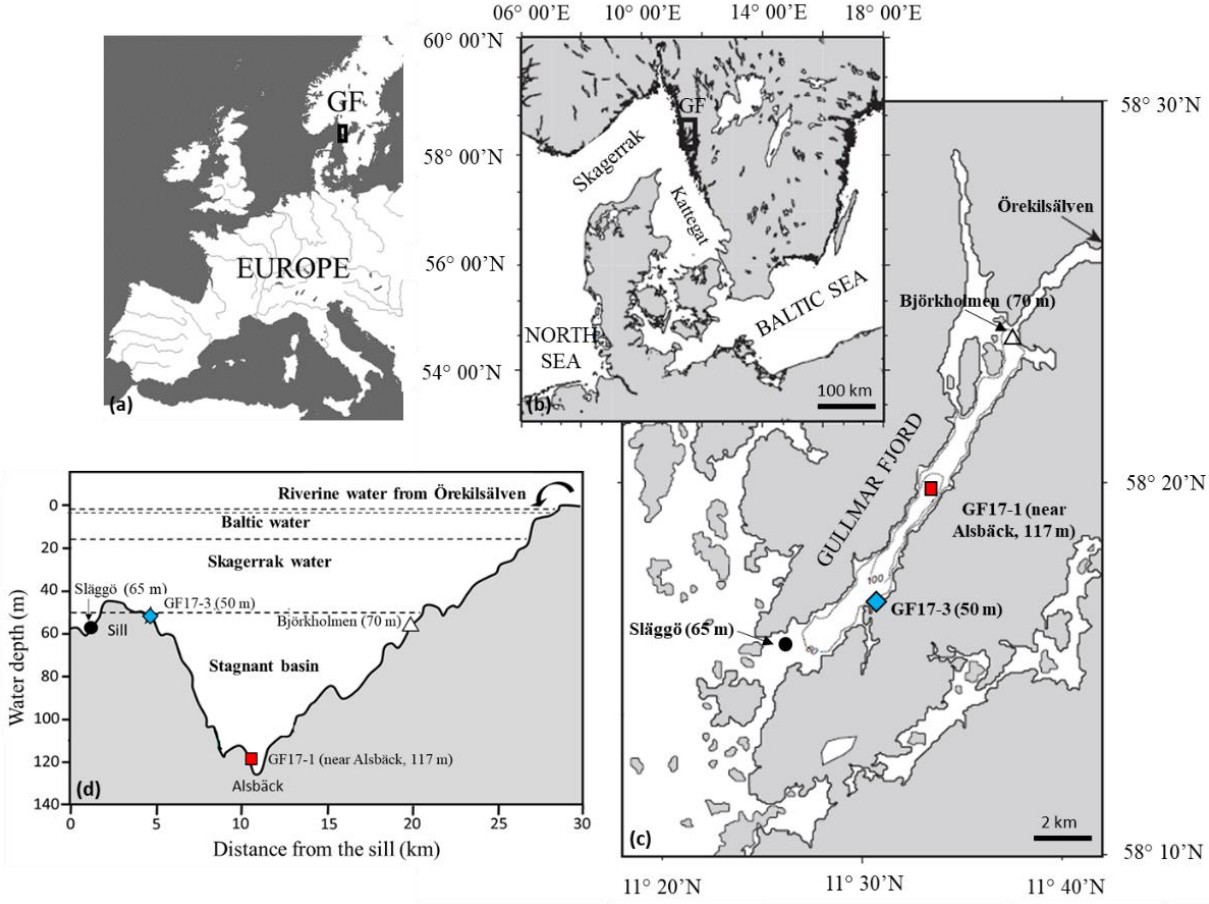

**Figure 2. (a-c) Location of studied stations in the Gullmar Fjord (Sweden); blue diamond: GF17-3 oxic station (50 m depth); red square: GF17-1 hypoxic station (117 m depth); dark circles: monitoring stations Släggö (65 m depth) and Björkholmen (70 m depth). (d) Transect from the sill with four Gullmar Fjord water masses and studied stations (modified from Arneborg et al., 2004).**



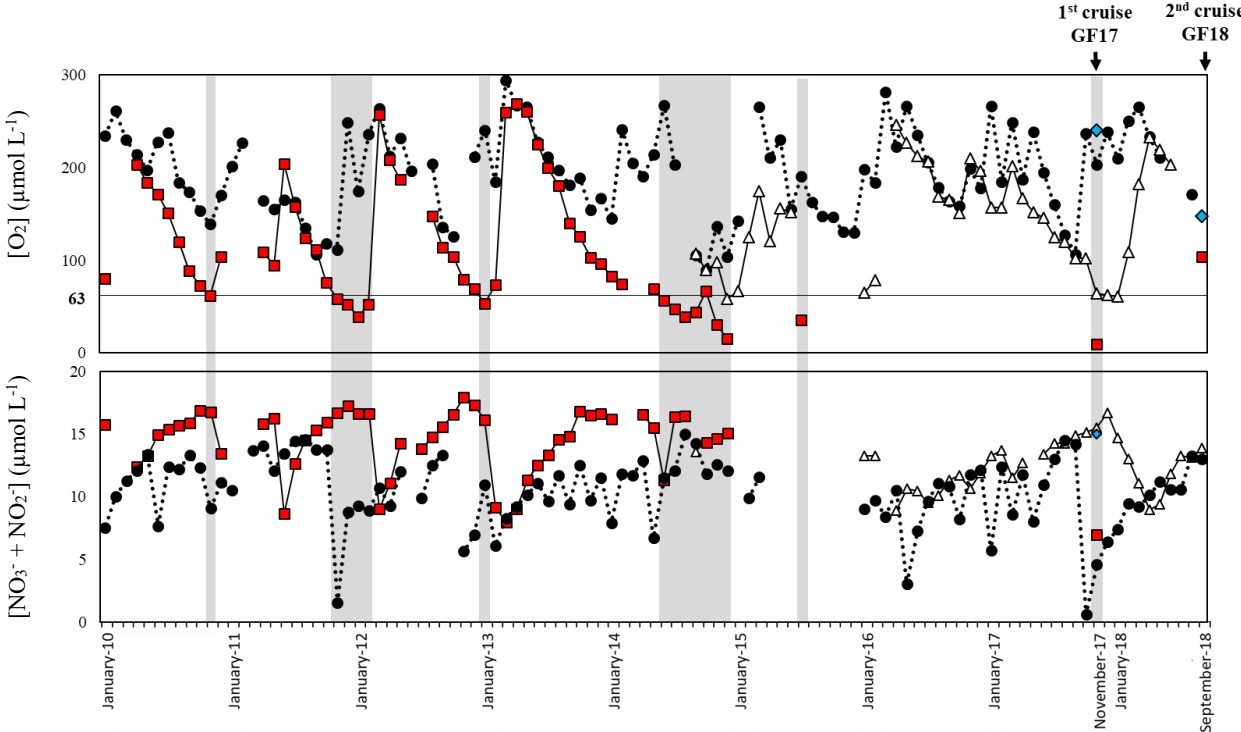

**Figure 3. Record from January 2010 to September 2018 of bottom water oxygen ([O₂]) and nitrite + nitrate ([NO₃⁻ +**

**NO₂⁻]) measurements from the monitoring stations Släggö (65 m depth; black dot), Björkholmen (70 m depth; white**

**triangle) and the sampling stations GF17-1 (Alsbäck, 117 m depth; red square) and GF17-3 (50 m depth; blue diamond).**

**The arrows indicate the date of the two sampling cruises: the first cruise GF17 (14th, 15th November 2017) and the**

**second cruise GF18 (5th September 2018). The grey zones indicate hypoxia threshold ([O2] < 63 µmol L⁻¹).**


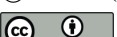




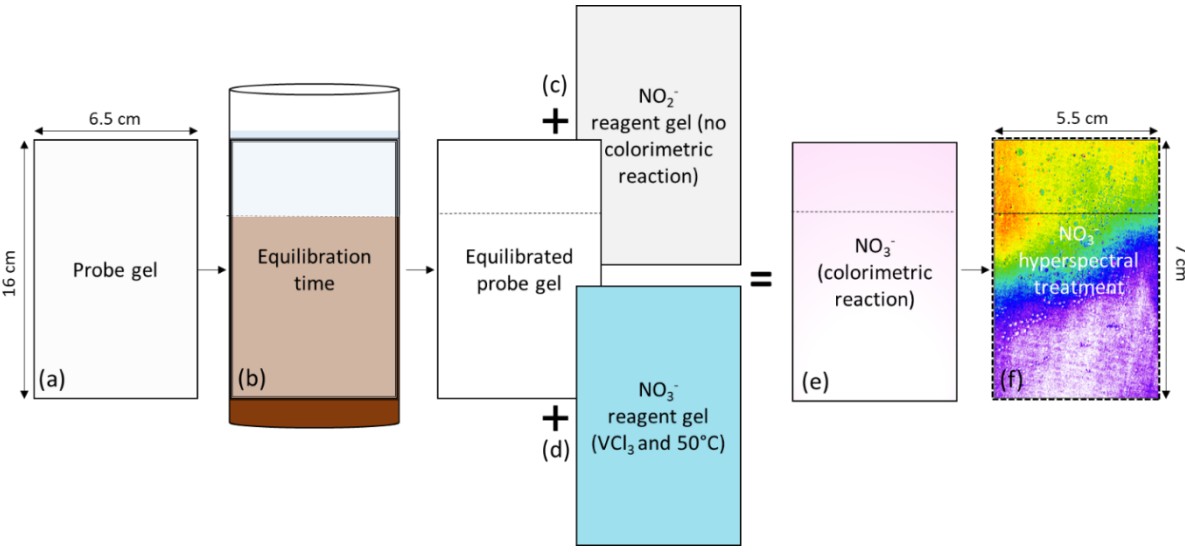

**Figure 4. Schematics of the nitrate 2D gel deployment and treatment. Details in the text.**




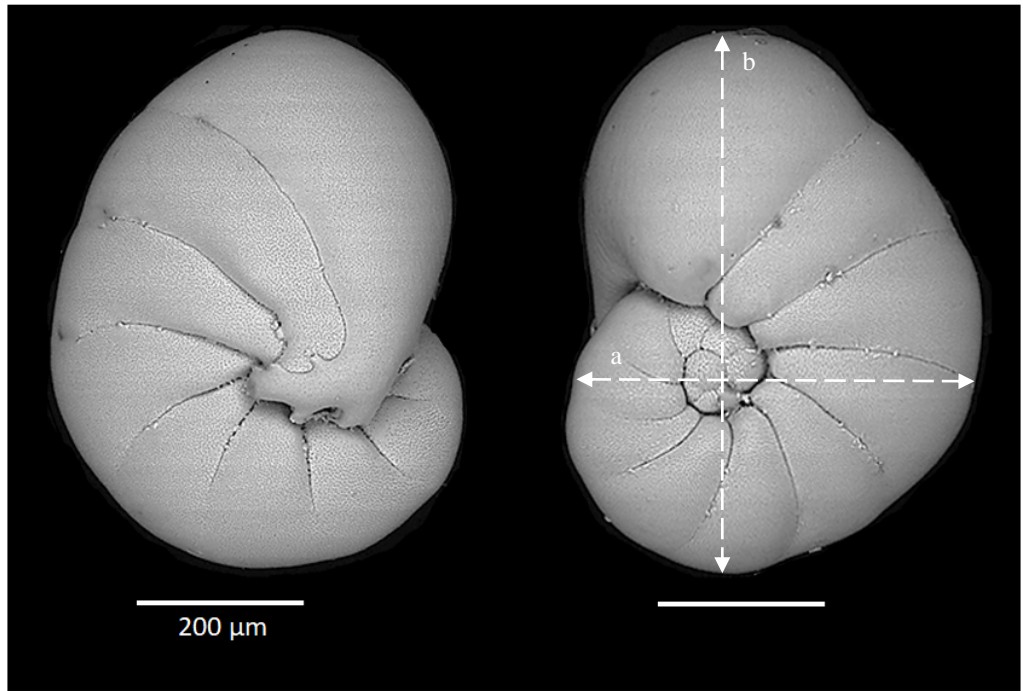

**Figure 5. Scanning Electronic Microscope images of a *Nonionella* sp. T1 from the GF17-3 oxic station in the Gullmar Fjord. White lines (a, b) correspond to measured distances serving for a spheroid prolate volume model.**





**Figure 6. Micro-distributions of living foraminifera densities in GF17-3 oxic station (a, b) and in GF17-1 hypoxic station (e, f).** *Nonionella* sp. T1 specimens are in black, the sum of the non-denitrifying species in grey colors and the small dots (e, f) show the other denitrifying species (known and potential candidates). The maps of porewater nitrate 2D gels are presented for stations GF17-3 (c) and GF17-1 (g). The sediment-water interface is represented by a black line at 0 cm depth (c, g) and the Oxygen Penetration Depth (OPD) is represented by the dashed line in bold at $4.7 \pm 0.2$ mm depth (c). Nitrate 1D profiles (d and h, black dots) are calculated using the average value of each pixel line of the nitrate distribution image (290 pixels wide), the standard deviation is represented by two fine dotted lines (c and g respectively). The corresponding best-fitting concentration profiles (red dots, d and h) and the production zones (red line) are modelled with PROFILE. The 1D profile corresponding to x = 1 mm (white line, g) is represented with a blue square profile (h) and the deep nitrate spot is indicated by a black arrow. The hatched grey zone (h) represents the detection limit of the nitrate 2D gel ($<1.7$ µmol L$^{-1}$).



**Table 1.** Summary of the invasive *Nonionella* sp. T1 contributions to benthic denitrification in the Gullmar Fjord. The porewater denitrifications zones come from PROFILE modelling (Fig. 6 d, h). To estimate the contributions of *Nonionella* sp.

T1 the counted specimens per zones was used. Two different approaches were used to estimate the contribution of *Nonionella* sp. T1: (A) divided the *Nonionella* sp. T1 denitrification rate by the nitrate porewater denitrification rate estimated from PROFILE modelling, then the second approach (B) divided the *Nonionella* sp. T1 denitrification rate by the denitrification rate from PROFILE plus the *Nonionella* sp. T1 denitrification rate. The calculations are detailed in Equation S2.

| Stations | Sediment depth interval of denitrification (cm) | *Nonionella* sp. T1 (counted specimens per zone) | Nitrate porewater denitrification rates ($\mu$mol m$^{-2}$ d$^{-1}$) | *Nonionella* sp. T1 denitrification rates ($\mu$mol m$^{-2}$ d$^{-1}$) | *Nonionella* sp. T1 contribution (%), approach A | *Nonionella* sp. T1 contribution (%), approach B |
|---|---|---|---|---|---|---|
| GF17-3A | 1.2 to 5 | 841 | 3.52 10$^{-07}$ | 1.63 10$^{-07}$ | 46 | 32 |
| GF17-3C | 1.2 to 5 | 1807 | 3.52 10$^{-07}$ | 3.51 10$^{-07}$ | 100 | 50 |
| GF17-1A | 0 to 1.6 | 3 | 2.34 10$^{-07}$ | 5.80 10$^{-10}$ | 0 | 0 |
| GF17-1C | 0 to 1.6 | 12 | 2.34 10$^{-07}$ | 2.32 10$^{-09}$ | 1 | 0 |






**Team list**

Constance Choquel

UMR 6112 LPG BIAF, Univ. Angers, Univ. Nantes, CNRS, France

constance.choquel@gmail.com

Emmanuelle Geslin

UMR 6112 LPG BIAF, Univ. Angers, Univ. Nantes, CNRS, France

emmanuelle.geslin@univ-angers.fr

Edouard Metzger

UMR 6112 LPG BIAF, Univ. Angers, Univ. Nantes, CNRS, France

edouard.metzger@univ-angers.fr

Helena L. Filipsson

Department of Geology, Lund University, Sweden

Helena.Filipsson@geol.lu.se

Nils Risgaard-Petersen

Department of Geosciences, Aarhus University, Denmark

nils.risgaard-petersen@bios.au.dk

Patrick Launeau

UMR 6112 LPG BIAF, Univ. Angers, Univ. Nantes, CNRS, France

patrick.launeau@univ-nantes.fr

Manuel Giraud

UMR 6112 LPG BIAF, Univ. Angers, Univ. Nantes, CNRS, France

Manuel.Giraud@univ-nantes.fr

Thierry Jauffrais

Ifremer, IRD, Univ. Nouvelle-Calédonie, Univ. La Réunion, CNRS,  UMR 9220 ENTROPIE, New Caledonia

Thierry.Jauffrais@ifremer.fr

Bruno Jesus

Université de Nantes, Mer Molécules Santé, EA 2160, France

bruno.jesus@univ-nantes.fr

Aurélia Mouret

UMR 6112 LPG BIAF, Univ. Angers, Univ. Nantes, CNRS, France

aurelia.mouret@univ-angers.fr

**Biogeosciences** Open Access
Discussions
EGU

## Author contributions

C.C. participated in the sampling cruise, did the foraminifera taxonomy, contributed to 2D gel experiments and analyses by hyperspectral camera. C.C. did the nitrate and oxygen respiration measurements. CC wrote the present manuscript. E.G. participated in the sampling cruise, contributed to foraminifera analysis, scientific discussions. E.M. participated in the sampling cruise, managed with A.M. the 2D gels experiments, and contributed to hyperspectral camera treatments and scientific discussions and manuscript rewriting. H.L.F managed with A.M the sampling cruise. H.L.F contributed to foraminifera taxonomy and scientific discussions and manuscript rewriting. N.R.P. managed the oxygen and nitrate respiration measurements and contributed to the scientific discussions. P.L. managed hyperspectral treatments for 2D gels and contributed to scientific discussion. M.G. participated in the 2D gel lab experiments and hyperspectral treatments. T.J. participated to the sampling cruise, contributed to 2D gels experiments and scientific discussions and manuscript rewriting. B.J. contributed to scientific discussion and manuscript rewriting. A.M. managed the sampling cruise and 2D gels experiments. A.M. contributed to hyperspectral camera treatments and scientific discussions and manuscript rewriting.

**Author information**

**Corresponding author**

Phone +33(0)2 41 73 53 82; fax: +33(0)2 41 73 53 52; e-mail: constance.choquel@gmail.com

## Competing interests

The authors declare no competing interest.

## Acknowledgements

The authors gratefully acknowledge the crews of the R/V Skagerak and Oscar von Sydow and the Kristineberg Marine Research Station, the hydrographic data used in the project are from SMHI's database – SHARK. The collection of data for SHARK is organized by the Swedish environmental monitoring program and funded by the Swedish Agency for Marine and Water Management (SWAM). Charlotte LeKieffre who helped during the sampling, the SCIAM (Service Commun d'Imagerie et d'Analyses Microscopiques) of Angers University for the SEM images. HLF acknowledges funding from the Swedish





Research Council VR (grant number 2017-04190). This project was funded by the French National Program MANGA-2D

(CNRS-INSU) and by the FRESCO project supported by the Region Pays de la Loire and by University of Angers.

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
