# Peer review of "Denitrification by benthic foraminifera and their contribution to N-loss from a fjord environment"

_Biogeosciences, 2020_

## Short Comment (SC1) · 6 Aug 2020

Congratulations to the authors for an interesting paper that will help understanding better the ecology of the Non-Indigenous Species (NIS) Nonionella sp. T1 in Europe. It is an important contribution to further evaluate the potential of this species to become invasive.

I have a small concern I would recommend the authors to address. Terminology and semantic is very important in science. Very often, there is a confusion between "introduced" and "invasive" status of NIS species. At the moment, it is well reported in Deldicq et al 2019 and also in Polovodova Asteman and Schönfeld 2015, the status of the NIS species Nonionella sp. T1 is still "introduced", she is not yet considered "inva-

sive" (although you may get confused by the term "invasion" in the title of Polovodova Asteman and Schönfeld). I suggest you edited the status from "invasive" to "Non-indigenous species (NIS)".

Last section of Deldicq et al 2019 rose the following question "May Nonionella sp. T1 become invasive"? We were not yet able to answer. Your contribution further shows that this species actually have the potential to become invasive, the high relative abundances confirming that its status may actually be considered as "invasive" in the Gullmar fjord.

---

## Short Comment (SC2) · 24 Aug 2020

Dear Vincent Bouchet, Thank you for your constructive comment and your interest in our work. I agree that consensus about the ecological status of Nonionella sp. T1 in Europe fjord needs to be reached.

In the previous literature three papers should be taken into account. The study of Polovodova Asteman and Schönfeld (2016) revealed the introduction of Nonionella stella in the North Sea and used the terminology "invasion" but they did not discuss this species as "invasive". Then, Nonionella stella was mentioned as "invasive" species by Charrieau et al. (2018), in the North Sea. Recently, your group (Deldicq et al., 2019), named it Nonionella sp. T1 after DNA analyses and assigned to it the status of intro-

Printer-friendly version / Discussion paper buttons.

[Figure]

duced Non-Indigenous Species (NIS) in the North Sea and discussed if Nonionella sp. T1 would become an invasive species.

The main aim of our study was not to show the invasion of Nonionella sp. T1, but the role of denitrifying species of benthic foraminifers in the geochemical cycle of nitrogen in the context of coastal hypoxia. However, we were surprised by its high relative density at the entrance of the Gullmar fjord and therefore by the potential contribution of this species to the mitigation of nitrogen through denitrification in the Gullmar Fjord that accounts to more than 50% of the benthic denitrification. The trend drawn by previous studies that our study seems to confirm definitely pleads to a change of the status of Nonionella sp. T1 from NIS to invasive. This is why the title of the manuscript seems appropriate even though some development could be done in the introduction and the discussion of the BGD manuscript for final publication in BG.

---

## Referee Comment (RC1) · Anonymous Referee #1 · 31 Aug 2020

This study is an important work using advanced techniques to better our understanding of the role of benthic foraminifera in the marine N cycle. In recent years, an increasing number of studies help us to see the "so far underestimated" contribution of benthic denitrification in N budget and cycling, and this study is an important input for the scientific advancement of this specific topic. Maps and figures are really nicely prepared and representative enough. I few questions and suggestions for the authors.

1) The title in the current format gives the impression of a study that only focuses on this invasive species in randomly chosen marine settings. Why exactly is important that this species is invasive to Gullmar Fjord? Overall, this study presents the important contribution of Nonionella sp. T1 regardless it is invasive or not and this is a really important input regarding our understanding of benthic N-Loss in such environments.

In my opinion, the title should include the regional characteristics including Gullmar Fjord or the North Sea rather than a generalized focus on invasive species' contribution to nitrate uptake. Or the overall discussion of this MS should include more of; what does this mean? This invasive species is increasing in numbers in the region (maybe in other areas too?) which is capable of such contribution to N dynamics and we are expecting to see ... in the future. The observation of its increase in the region is valuable. Nevertheless, I am not sure this is exactly the message of this specific study.

2) Do authors think before the invasion of Nonionella sp. T1 benthic denitrification was overall less than their observations in this study or it has been overall the same values, but the other species are simply losing the competition now in the region? Is there any indication or previous study focusing on that? if this is the first time observation on this specific topic in this region, the authors should emphasize it even more.

3) Please provide references for benthic foraminifera taxonomy in supplementary material, considering which publication (maybe even which figure) was used for identification of the species listed in Table S3 and S4.

Abstract: Line 14: there is no flow/connection between the first 2-3 sentences. It would be better to focus on first the importance of invasive species in certain regions or the importance of oxygen, nitrate dynamics in such regions. I think authors should decide how to formulate the most important message of this MS. Line 18: micro-distribution... microhabitat instead?. Line 19: worth to mention Gel methodology already here for least confusion of 2D geochemistry concept. The next sentence also needs a reshape giving a broader idea of these contrasting sites. Oxygenated overlying and bottom waters with high nitrate content in porewaters vs hypoxic bottom waters where porewater is nitrate scarce.

Introduction: First sentence: I am confused with nomenclature, unit choice, and conversion of values here. There are many studies focusing on different values for the term hypoxia so I highly recommend citing the publication that the authors followed.

This is also valid for unit choice, I am familiar with dissolved oxygen concentration units of mL/L and umol/kg or umol/L. Generally, 2 ml/l is circa 90 umol/l. Most of the studies concerning benthic foraminifera in low oxygen environments focus on these units. I just wonder which study the authors decided to follow in this case.

Line 33: . . .contrasted dissolved O2 conditions: Over what time interval? a year? different seasons? or different sampling sites? I know this information will be mentioned later but it would be nice to give the information here already.

Line 44: "total denitrification". Overall, denitrification together with anammox is also called N-loss. I recommend authors have a look at some other reviews on marine N cycle: Galloway et al., 2004, Gruber and Sarmiento, 1997; Gruber and Galloway, 2008. Maybe even Sigman et al 2009 (is in the direction of N isotope chemistry but is a nice review). These are reviews that would give a bit more insight and overview of the marine N cycle with perspective to open sea/ocean. There are many publications on coastal systems and while investigation on N2 loss and its impact on eutrophication I came across to Seitzinger 1988 I think should be included either to the introduction or the discussion to make the findings of this study more pronounced. It is worth mentioning the potential benefit of benthic denitrification to eutrophication already in the introduction giving examples from previous studies.

Line 48: . . . nitrification cannot process under low oxygen conditions. How low? Please indicate the values here.

Section 2 Methods Suggestion for site or expedition indicator throughout the text: Instead of 1st and 2nd cruise, authors could use years, e.g., 2017 and 2018.

Line 109: (see previous studies) please indicate references instead.

Line 127: is there a special reason for the choice of 100 um fraction? Whereas well-accepted fractions are 63, 125, and 150 um?

Line 140 and figure 4: Is Figure 4 needed? Is this method described here the first time

and different from Metzger et al., 2016?

Line 202: I find Table S1 rather important for this MS. What about involving it to the main MS but not only in supplementary information?

4. Discussion: Line 301: I think it should be GF17-1A and 1C in the parenthesis.

Line 309: (our results) data not shown and presented? If so, please mention or indicate where this information comes from. In the same line, it would be better to mention some of the previous studies showing differences too.

I recommend changing the titles for the section 4.2 and 4.3 to "...T1/foraminifera habitat in relation with the nitrate micro-distribution..." since there might be other factors having an impact on the ecology of these species, it would be better to keep the focus on nitrate and oxygen in these sections of the discussion.

Line 395: once again discussion on benthic N loss contribution to eutrophication: I think this needs a broader discussion and requires some references. Moreover, does N2 flux from sediment promote N2 fixation, and thus, e.g., cyanobacterial activity? are there studies focusing on N2 fix vs N loss in Gullmar Fjord or similar settings? I think considering these would improve the discussion significantly.

---

## Referee Comment (RC2) · Anonymous Referee #2 · 17 Sep 2020

This study points out the importance of benthic foraminifera in the marine nitrogen cycle. Denitrification and oxygen respiration of the abundant and non-indigenous species Nonionella sp. T1 in the Gullmar Fjord were measured. Further, a state-of-the-art method was applied to compare the distributions of benthic foraminifera and the availability of sedimentary nitrate. Additionally, the contribution of Nonionella sp. T1 to the benthic denitrification was quantified and revealed, that this species has a strong impact on the nitrogen cycle at an oxic station at the Swedish Gullmar Fjord. I have two points to consider:

1)The title of the study implies, that the work focuses on total nitrate uptake of a specific benthic foraminifer. However, the emphasis of the first part in the discussion of this study implies a thorough taxonomic investigation of the Fjord fauna, which is not

the case in this study. I agree with the authors, that there is an ongoing change in the benthic foraminiferal community of the Gullmar Fjord. But to verify this trend and to discuss its consequences, a longer-term monitoring study observing seasonal fluctuations of the benthic foraminiferal community together with environmental parameters at several stations within the fjord is necessary. Further, a more detailed comparison with previous literature would be necessary. I think the authors should point out, that such monitoring studies (including the 63 - 125$\mu$m size fraction) are important for the future, specifically considering the new observations of this study.

2)I agree with the author of the short comment considering the invasive status of Nonionella sp. T1. Certainly, this species is proven to be non-indigenous. However, the actual invasive status of this species is not proven yet. It is not yet clear, if the occurrence of Nonionella sp. T1 is responsible for the disappearance of any other species in the Fjord, nor is there any evidence, that this species is harmful for the ecosystem of the Gullmar Fjord. On the contrary, the authors point out, that this species might even be of advantage for the trophic status of the fjord. It is important to stick with correct ecological terminology to avoid confusion in further research. I would recommend to change the term 'invasive' to 'non-indigenous'.

Additionally, I would like to add a few technical corrections and minor remarks:

Introduction:

Line 29: 'and thereby to survive' should be 'and thereby survive'

Line 32: 'This study focus on...' should be 'This study focuses on...'

Material and Methods:

Line 127: 'Fixed samples were sieved and the > 100 $\mu$m fraction was examined...' Did you remove any larger meiofauna e.g. by sieving through a larger sieve (5 mm, 2 mm, 1mm)? If so, this should be mentioned too, since adults of larger denitrifying genera e.g. Globobulimina often cannot pass through a 1 mm sieve.

Discussion:

Line 292: I would consider to change the title of this section into something like: 'Abundance of Nonionella sp. T1 in comparison with other species'

Line 315: I think there is something a little bit wrong with this sentence. Should it be something like: 'That the foraminiferal fauna described in the present study differs, is the consequence...'

Line 327: Did Polodova Asteman and Schönfeld (2015) sample the same location at the oxic part of the fjord?

Line 359: Could propagules also be a reason for the survival or re-appearance of the non-denitrifying species in the hypoxic part of the fjord?

Line 392: I would be careful with this consideration, because other well oxygenated areas of the Fjord might be dominated by other species - depending on depth or other environmental parameters.

Figures:

Figure 6: It should be 'Depth (mm)' for GF17-3A and 3C and GF17-1A and 1C and not Depth (cm).

---

## Author Comment (AC1) · 5 Oct 2020

Constance Choquel on behalf of all authors

constance.choquel@gmail.com

Dear Referee,

Thank you for your constructive comment and your interest in our work. We agree with the majority of the suggestions that you bring to our study. Indeed, this study highlights the importance of the contributions of denitrifying benthic foraminifera regarding contrasted oxygen and nitrate conditions at two different sites in a fjord. We agree that the introductory part on the nitrogen cycle deserves to be better contextualized with more general bibliographical references. It is, however, premature to engage in the description of the effects of foram denitrification on primary production and nitrogen fixation. We prefer to remain cautious in the discussion and not speculate too much.

Question 1: In my opinion, the title should include the regional characteristics including Gullmar Fjord or the North Sea rather than a generalized focus on invasive species' contribution to nitrate uptake. Or the overall discussion of this MS should include more of; what does this mean? This invasive species is increasing in numbers in the region (maybe in other areas too?) which is capable of such contribution to N dynamics and we are expecting to see in the future. The observation of its increase in the region is valuable. Nevertheless, I am not sure this is exactly the message of this specific study.

Answer 1: We suggest a novel title as "Total nitrate uptake by benthic foraminifera in a sill fjord environment"

Question 2: Do authors think before the invasion of Nonionella sp. T1 benthic denitrification was overall less than their observations in this study or it has been overall the same values, but the other species are simply losing the competition now in the region? Is there any indication or previous study focusing on that? if this is the first time observation on this specific topic in this region, the authors should emphasize it even more.

Answer 2: Station GF17-3 (50 m) was sampled for the first time in this study. There is therefore no records on the benthic denitrification and the assemblages of foraminifera at this precise location of the Fjord.

Question 3: Please provide references for benthic foraminifera taxonomy in supplementary material, considering which publication (maybe even which figure) was used for identification of the species listed in Table S3 and S4.
Answer 3: Yes we will add them. We used Charrieau et al., (2018) and references therein.

Question 4: Abstract: Line 14: there is no flow/connection between the first 2-3 sentences. It would be better to focus on first the importance of invasive species in certain regions or the importance of oxygen, nitrate dynamics in such regions. I think authors should decide how to formulate the most important message of this MS. Line 18: micro-distribution: microhabitat instead? Line 19: worth to mention Gel methodology already here for least confusion of 2D geochemistry concept. The next sentence also needs a reshape giving a broader idea of these contrasting sites. Oxygenated overlying and bottom waters with high nitrate content in porewaters vs hypoxic bottom waters where porewater is nitrate scarce.

Answer 4: We will focus the abstract on the importance of the contribution of denitrifying forams in the Gullmar fjord according to the contrasting geochemical conditions in oxygen and nitrates. We will rewrite the sentence concerning the methodology of 2D gels.

Question 5: Introduction: First sentence: I am confused with nomenclature, unit choice, and conversion of values here. There are many studies focusing on different values for the term hypoxia so I highly recommend citing the publication that the authors followed. This is also valid for unit choice, I am familiar with dissolved oxygen concentration units of mL/L and umol/kg or umol/L. Generally, 2 ml/l is circa 90 umol/l. Most of the studies concerning benthic foraminifera in low oxygen environments focus on these units. I just wonder which study the authors decided to follow in this case.

Answer 5: We use only the unit µmol/L in the study. The hypoxia threshold used is 63 µmol/L cited by Breitburg 2018 corresponding to 2 mg/L and 1.4 ml/L.

Question 6: Line 33: contrasted dissolved O2 conditions: Over what time interval? a year? Different seasons? or different sampling sites? I know this information will be mentioned later but it would be nice to give the information here already.

Answer 6 : Yes. Two contrasting oxygen stations, one hypoxic in the deep basin (GF17-1) at the end of autumn 2017 and an oxic station towards the mouth of the fjord.

Question 7: Line 44: "total denitrification". Overall, denitrification together with anammox is also called N-loss. I recommend authors have a look at some other reviews on marine N cycle: Galloway et al., 2004, Gruber and Sarmiento, 1997; Gruber and Galloway, 2008. Maybe even Sigman et al 2009 (is in the direction of N isotope chemistry but is a nice review). These are reviews that would give a bit more insight and overview of the marine N cycle with perspective to open sea/ocean. There are many publications on coastal systems and while investigation on N2 loss and its impact on eutrophication I came across to Seitzinger 1988 I think should be included either to the introduction or the discussion to make the findings of this study more pronounced. It is worth mentioning the potential benefit of benthic denitrification to eutrophication already in the introduction giving examples from previous studies.

Answer 7: Some of these references can support the introductory part of the state of the art of the nitrogen cycle in marine sediments in order to contextualize more broadly the importance of identifying the sources and outputs of nitrogen from a system (Galloway, 2004). In most coastal environments such as the Baltic Sea the loss of nitrogen through denitrification exceeds the supply of nitrogen through nitrogen fixation. These sink regions of the ocean are the areas associated with the anoxic regions (Grubber and Sarmiento 1997). When benthic denitrification exceeds nitrogen fixation, eutrophication can be mitigated via nitrogen loss (Seitzinger 1988). The Gullmar Fjord would be a sink region.

In the last part of the discussion (4.4) it is possible to briefly provide more details on the eutrophication state of Gullmar Fjord. Primary production in Gullmar Fjord is dominated by diatoms bloom in spring and autumn (Lindahl and Hernroth, 1983). Since the 1990s Lindahl et al. (2003) observed the increase in primary production of the Gullmar Fjord, therefore a potential eutrophication of the Fjord. This increase in original productivity also shown in the adjacent Kattegat could be related to the nitrogen input loading from the land and atmosphere (Carstensen et al. (2003)). Lindahl et al. (2003), argued that primary production of the Gullmar fjord was due to climatic forces resulting from a strong positive North Atlantic Oscillation (NAO) index, which increased the availability of deepwater nutrients

(Kattegat nitrate-rich) and due to warmer ocean. The benthic denitrification of Gullmar Fjord makes it possible not to supply the system with nitrogen available for primary producers. Denitrifying foraminifera including *Nonionella* sp. T1 could thus help counterbalance this eutrophication by increasing the loss of $N_2$. Glock et al., (2013) also supported denitrifying forams in OMZ contributed to N-loss (until 46%). Then, foraminifera intracellular nitrates become unavailable to the system and can be bio-transported and permanently sequestered in sediments (Glock et al., 2013; Prokopenko et al., 2011).

Question 8: Line 48: nitrification cannot process under low oxygen conditions. How low? Please indicate the values here.

Answer 8: According to Mortimer et al., (2004) and Rysgaard et al., (1994) once the oxygen in the sediment is no longer detected (close to 0 µmol / L) the nitrification also becomes undetectable.

Question 9: Section 2 Methods Suggestion for site or expedition indicator throughout the text: Instead of 1st and 2nd cruise, authors could use years, e.g., 2017 and 2018.

Answer 9: Yes indeed it may be clearer using the dates of the missions

Question 10: Line 109: (see previous studies) please indicate references instead.

Answer 10: Nordberg and al., 2000; Filipsson and Nordberg, (2004)

Question 11: Line 127: is there a special reason for the choice of 100 um fraction? Whereas well accepted fractions are 63, 125, and 150 um?

Answer 11: In the previous studies in the Gullmar Fjord, Skagerrak and Kattegat, the size fraction > 100 µm has most commonly been used for foraminiferal analyses (see Charrieau et al., 2018 and references therein).

Question 12: Line 140 and figure 4: Is Figure 4 needed? Is this method described here the first time and different from Metzger et al., 2016?

Answer 12: This is the same method as Metzger et al., 2016 but since the steps in this method can be difficult to follow for non-specialists we find the diagram helps to easily visualize the method.

Question 13: Line 202: I find Table S1 rather important for this MS. What about involving it to the main MS but not only in supplementary information?

Answer 13: This table is better in SI and it is easy accessible on the webpage.

Question 14: 4. Discussion: Line 301: I think it should be GF17-1A and 1C in the parenthesis.

Answer 14: ok

Question 15: Line 309: (our results) data not shown and presented? If so, please mention or indicate where this information comes from. In the same line, it would be better to mention some of the previous studies showing differences too.

Answer 15: Ok

Question 16: I recommend changing the titles for the section 4.2 and 4.3 to ": : :T1/foraminifera habitat in relation with the nitrate micro-distribution: : :" since there might be other factors having an impact on the ecology of these species, it would be better to keep the focus on nitrate and oxygen in these sections of the discussion.

Answer 16: We will merge the two parts 4.2 and 4.3

4.3 The foraminifera ecology considering the nitrate micro-distribution

Inside, there will be a first paragraph about oxic station and a second paragraph about hypoxic station.

Question 17: Line 395: once again discussion on benthic N loss contribution to eutrophication: I think this needs a broader discussion and requires some references. Moreover, does N2 flux from sediment promote N2 fixation, and thus, e.g., cyanobacterial activity? Are there studies focusing on N2 fix vs N loss in Gullmar Fjord or similar settings? I think considering these would improve the discussion significantly.

Answer 17: it's difficult to answer this question without getting too speculative

The question here suggests that nitrogen supply via benthic denitrification of the forams could be captured by $N_2$-fixing cyanobacteria and participate in their development. Significant cyanobacteria blooms are already known in the Baltic Sea (Boesch 2003 Swedish agency report). In the Gullmar fjord there are few studies on cyanobacteria (Croot, 2003) the evolution of $N_2$-fixation by these cyanobacteria in Gullmar Fjord is not obvious and there is a lack of data. Benthic denitrification of the forams may participate in the N pool to be fixed by cyanobacteria but this hypothesis is too speculative, cyanobacteria in Gullmar Fjord do not appear to be a major threat to the system at this time.

---

## Author Comment (AC2) · 5 Oct 2020

Constance Choquel behalf on all the authors.

constance.choquel@gmail.com

Dear Referee,

Thank you for your constructive comment and your interest in our work. We agree with the majority of the suggestions that you bring to our study. The status of *Nonionella* sp T1 remains unclear. We are to follow the recommendations made by V. Bouchet by introducing *Nonionella* sp. T1 as Non-Indigenous Species then, in discussion we will discuss its invasiveness in Gullmar Fjord. Indeed, the dominance of *Nonionella* sp. T1 could be harmful for the foraminifera diversity. I am aware that this study must be followed by a long bio-monitoring > 63 µm (seasonal, different depths stations) to validate the ongoing change in Gullmar Fjord fauna.

Question 1: The title of the study implies, that the work focuses on total nitrate uptake of a specific benthic foraminifer. However, the emphasis of the first part in the discussion of this study implies a thorough taxonomic investigation of the Fjord fauna, which is not the case in this study. I agree with the authors, that there is an ongoing change in the benthic foraminiferal community of the Gullmar Fjord. But to verify this trend and to discuss its consequences, a longer-term monitoring study observing seasonal fluctuations of the benthic foraminiferal community together with environmental parameters at several stations within the fjord is necessary. Further, a more detailed comparison with previous literature would be necessary. I think the authors should point out, that such monitoring studies (including the 63 – 125µm size fraction) are important for the future, specifically considering the new observations of this study.

Answer 1: We agree that a long monitoring would be necessary to validate the change in fauna and include a study with a smaller fraction.

Question 2: I agree with the author of the short comment considering the invasive status of *Nonionella* sp. T1. Certainly, this species is proven to be non-indigenous. However, the actual invasive status of this species is not proven yet. It is not yet clear, if the occurrence of Nonionella sp. T1 is responsible for the disappearance of any other species in the Fjord, nor is there any evidence, that this species is harmful for the ecosystem of the Gullmar Fjord. On the contrary, the authors point out, that this species might even be of advantage for the trophic status of the fjord. It is important to stick with correct ecological terminology to avoid confusion in further research. I would recommend to change the term 'invasive' to 'non-indigenous'.

Answer 2: We agree with V. Bouchet comment. I will introduce *Nonionella* sp. T1 as a Non-Indigenous Species (Deldick et al., 2019). Then, in the discussion We will mention the invasive character of this species in the Gullmar Fjord in view of its strong increase in density at the entrance to the Fjord. There is no evidence yet that *Nonionella* sp. T1 can harm the ecosystem, however *Nonionella* sp. T1 could affect the fauna of foraminifera. Indeed, the specific richness (S) and the Shannon index (H) decrease with sediment depth sediment in the GF17-3 station while the dominance (D) due to *Nonionella* sp. T1 increases (see graphs GF17-3A and 3C). In the hypoxic station (GF17-1), the dominance is driven by *Cassidulina laevigata* and *Bulimina marginata* which dominated the fauna.

Specific richness

Shannon Index

Dominance

Specific richness

Shannon Index

Dominance

Specific richness

Shannon Index

Dominance

Specific richness

Shannon Index

Dominance

Additionally, I would like to add a few technical corrections and minor remarks:

Introduction:

Question 3: Line 29: 'and thereby to survive' should be 'and thereby survive'
Answer 3: ok

Question 4: Line 32: 'This study focus on...' should be 'This study focuses on...'
Answer 4: Ok

Material and Methods:

Question 5: Line 127: 'Fixed samples were sieved and the > 100 m fraction was examined...' Did you remove any larger meiofauna e.g. by sieving through a larger sieve (5 mm, 2 mm, 1mm)? If so, this should be mentioned too, since adults of larger denitrifying genera e.g. *Globobulimina* often cannot pass through a 1 mm sieve.

Answer 5: the sieves used are

| >355 | 355-150 | 150-125 | 125-100 |
|------|---------|---------|---------|

No 1 mm sieve was used there should be no loss of *Globobulimina*.

Discussion:

Question 6: Line 292: I would consider to change the title of this section into something like: 'Abundance of Nonionella sp. T1 in comparison with other species'

Answer 6: yes we will change the title to be more careful about the change of fauna.

Question 7: Line 315: I think there is something a little bit wrong with this sentence. Should it be something like: 'That the foraminiferal fauna described in the present study differs, is the consequence...'
Answer 7: We will rewrite this sentence.

Question 8: Line 327: Did Polodova Asteman and Schönfeld (2015) sample the same location at the oxic part of the fjord?

Answer 8: No, they sampled in the deep Alsbäck station which was oxic at the time of the sampling in August 2013 and July 2014. They sampled a station in the Skagerrak near the mouth of the fjord in June 2013, we compared my oxic station with this data out of the fjord.

Question 9: Line 359: Could propagules also be a reason for the survival or re-appearance of the non-denitrifying species in the hypoxic part of the fjord?

Answer 9 : Yes, propagules can disperse and reproduce when environmental conditions are favourable (Alve and Goldstein, 2003). However, there is no change in density of *Nonionella* sp. T1 at the Alsbäck station from the densities found by Polovodova Asteman and Schönfeld (2015). It would be interesting to look again at this Alsbäck station to see if there is an evolution of the densities of *Nonionella* sp. T1 and if there is a seasonality of denitrifying foraminifera depending on the oxygenation conditions (hypoxic vs oxic).

Question 10: Line 392: I would be careful with this consideration, because other well oxygenated areas of the Fjord might be dominated by other species - depending on depth or other environmental parameters.

Answer 10: Yes to test this hypothesis it would be necessary to sample several oxic stations at different depths in the fjord.

Question 11: Figure 6: It should be 'Depth (mm)' for GF17-3A and 3C and GF17-1A and 1C and not Depth (cm).

Answer 11: ok

---

## Author Response (AR1)

[revised manuscript text omitted]

constance.choquel@gmail.com on behalf of the coauthors

Dear Editor and referees,

Thank you for your interest in our work and your comments to improve the paper "Denitrification by
benthic foraminifera and their contribution to N-loss from a fjord environment". Please find the revised
manuscript attached. The main corrections performed to the manuscript are highlighted in yellow.

Briefly, we have changed the title of the manuscript, the new title highlights the denitrification of
foraminifera and their impact on the nitrogen cycle. The abstract has been adapted accordingly.
In the first paragraph of the introduction a contextualization of the importance of the nitrogen cycle in
semi-enclosed environments subject to hypoxia has been added. Then, a paragraph in discussion 4.3 has
been added to inform readers about the eutrophication state of Gullmar Fjord. Discussion sections
formerly 4.2 and 4.3 have been merged under the name " 4.2 Foraminifera ecology considering nitrate
micro-distribution".
To take into account the remarks of the short comment and the referees, we have changed the term
"invasive" *Nonionella* sp. T1 by non-indigenous species (NIS). The term "invasive" is introduced because
it is cited in the existing literature. The potential invasiveness of *Nonionella* sp. T1 in the Gullmar fjord
is mentioned later in  the discussion sub-section 4.1.
Figure 4 of the material and method about the 2D gel method has been removed as potential interested
readers can consult the original paper that details the procedure.
A conversion and a unit error have been found for the denitrification rates (nmol $cm^{-3}$ $s^{-1}$). The final
contribution results remain unchanged as the conversion error was done for both denitrification rates for
foraminifera and cores (see changes Fig. 5, Table 1, Annex Equation S2, and associated text).
For more details on minor changes please refer to  the replies to referees.

Best regards.

Constance Choquel
constance.choquel@gmail.com

Dear Referee 1,

Thank you for your constructive comment and your interest in our work. I agree with the majority of the suggestions that you bring to our study. Indeed, this study targeted the importance of the contributions of
denitrifying benthic foraminifera regarding contrasted oxygen and nitrate conditions at two different sites in the Gullmar Fjord. Indeed, the introductory part on the nitrogen cycle deserves to be better contextualized with more general bibliographical references. I think it is premature to engage in the description of the effects of denitrification of forams on primary production and nitrogen fixation. I prefer to remain cautious about the prospects of the discussion so as not to make too precise speculation.

Question 1: In my opinion, the title should include the regional characteristics including Gullmar Fjord or the North Sea rather than a generalized focus on invasive species' contribution to nitrate uptake. Or the overall discussion of this MS should include more of; what does this mean? This invasive species is increasing in numbers in the region (maybe in other areas too?) which is capable of such contribution to
N dynamics and we are expecting to see in the future. The observation of its increase in the region is valuable. Nevertheless, I am not sure this is exactly the message of this specific study.
Answer 1: I suggest a novel title as "Total nitrate uptake by benthic foraminifera in the Gullmar Fjord"

Question 2: Do authors think before the invasion of Nonionella sp. T1 benthic denitrification was overall
less than their observations in this study or it has been overall the same values, but the other species are simply losing the competition now in the region? Is there any indication or previous study focusing on that? if this is the first time observation on this specific topic in this region, the authors should emphasize it even more.
Answer 2: Station GF17-3 (50 m) was sampled for the first time in this study. There is therefore no
retreat on the benthic denitrification and the assemblages of foraminifera at this precise location of the Fjord.

Question 3: Please provide references for benthic foraminifera taxonomy in supplementary material, considering which publication (maybe even which figure) was used for identification of the species listed
in Table S3 and S4.
Answer 3: Yes I will add them. I looked at Charrieau et al., (2018)

Question 4: Abstract: Line 14: there is no flow/connection between the first 2-3 sentences. It would be better to focus on first the importance of invasive species in certain regions or the importance of oxygen,
nitrate dynamics in such regions. I think authors should decide how to formulate the most important message of this MS. Line 18: micro-distribution: microhabitat instead? Line 19: worth to mention Gel methodology already here for least confusion of 2D geochemistry concept. The next sentence also needs a reshape giving a broader idea of these contrasting sites. Oxygenated overlying and bottom waters with high nitrate content in porewaters vs hypoxic bottom waters where porewater is nitrate scarce.

Answer 4: I will focus the abstract on the importance of the contribution of denitrifying forams in the Gullmar fjord according to the contrasting geochemical conditions in oxygen and nitrates. Rewrite the sentence concerning the methodology of 2D gels.

Question 5: Introduction: First sentence: I am confused with nomenclature, unit choice, and conversion of values here. There are many studies focusing on different values for the term hypoxia so I highly recommend citing the publication that the authors followed. This is also valid for unit choice, I am familiar with dissolved oxygen concentration units of mL/L and umol/kg or umol/L. Generally, 2 ml/l is circa 90 umol/l. Most of the studies concerning benthic foraminifera in low oxygen environments focus on these 880 units. I just wonder which study the authors decided to follow in this case.

Answer 5: I use only the unit µmol/L in the study. The hypoxia threshold used is 63 µmol/L cited by Breitburg 2018.

Question 6: Line 33: contrasted dissolved O2 conditions: Over what time interval? a year? Different seasons? or different sampling sites? I know this information will be mentioned later but it would be nice to give the information here already.
Answer 6 : Yes, to be re-specified. Two contrasting oxygen stations, one hypoxic in the deep basin (GF17-1) at the end of autumn 2017 and an oxic station at the entrance to the fjord.

Question 7: Line 44: "total denitrification". Overall, denitrification together with anammox is also called N-loss. I recommend authors have a look at some other reviews on marine N cycle: Galloway et al., 2004, Gruber and Sarmiento, 1997; Gruber and Galloway, 2008. Maybe even Sigman et al 2009 (is in the direction of N isotope chemistry but is a nice review). These are reviews that would give a bit more insight 895 and overview of the marine N cycle with perspective to open sea/ocean. There are many publications on coastal systems and while investigation on N2 loss and its impact on eutrophication I came across to Seitzinger 1988 I think should be included either to the introduction or the discussion to make the findings of this study more pronounced. It is worth mentioning the potential benefit of benthic denitrification to eutrophication already in the introduction giving examples from previous studies.

Answer 7: Some of these references can support the introductory part of the state of the art of the nitrogen cycle in marine sediments in order to contextualize more broadly the importance of identifying the sources and outputs of nitrogen from a system (Galloway, 2004). In most coastal environments such as the Baltic Sea the loss of nitrogen through denitrification exceeds the supply of nitrogen through nitrogen fixation. 905 These sink regions of the ocean are the areas associated with the anoxic regions (Grubber and Sarmiento 1997). When benthic denitrification exceeds nitrogen fixation, eutrophication can be mitigated via nitrogen loss (Seitzinger 1988). The Gullmar Fjord would be a sink region.
In the last part of the discussion (4.4) it is possible to briefly provide more details on the eutrophication state of Gullmar Fjord. Primary production in Gullmar Fjord is dominated by diatoms bloom in spring 910 and autumn (Lindahl and Hernroth, 1983). Since the 1990s Lindahl et al. (2003) observed the increase in primary production of the Gullmar fjord, therefore a potential eutrophication of the Fjord. This increase in original productivity also shown in the adjacent Kattegat could be related to the nitrogen input loading from the land and atmosphere (Carstensen et al. (2003)). Lindahl et al. (2003), argued that primary production production of the Gullmar fjord was due to climatic forces resulting from a strong positive

North Atlantic Oscillation (NAO) index, which increased the availability of deepwater nutrients (Kattegat nitrate-rich) and due to warmer ocean surface. The benthic denitrification of Gullmar Fjord makes it possible not to supply the system with nitrogen available for primary producers. Denitrifying foraminifera including *Nonionella* sp. T1 could thus help counterbalance this eutrophication by increasing the loss of $N_2$. Glock et al., (2013) also supported denitrifying forams in OMZ contributed to N-loss (until 46%).

Then, foraminifera intracellular nitrates become unavailable to the system and can be bio-transported and permanently sequestered in sediments (Glock et al., 2013; Prokopenko et al., 2011).

Question 8: Line 48: nitrification cannot process under low oxygen conditions. How low? Please indicate the values here.

Answer 8: According to Mortimer et al., (2004). Once the oxygen in the sediment is no longer detected (close to 0 µmol / L) the nitrification also becomes undetectable.

Question 9: Section 2 Methods Suggestion for site or expedition indicator throughout the text: Instead of 1st and 2nd cruise, authors could use years, e.g., 2017 and 2018.

Answer 9: Yes indeed it may be clearer using the dates of the missions

Question 10: Line 109: (see previous studies) please indicate references instead.
Answer 10: Nordberg and al., 2000; Filipsson and Nordberg, (2004)

Question 11: Line 127: is there a special reason for the choice of 100 um fraction? Whereas well accepted fractions are 63, 125, and 150 um?
Answer 11: In the previous studies in the Gullmar Fjord, the size fraction > 100 µm has most commonly been used for foraminiferal analyses (see Charrieau et al., 2018).

Question 12: Line 140 and figure 4: Is Figure 4 needed? Is this method described here the first time and different from Metzger et al., 2016?
Answer 12: This is the same method as Metzger et al., 2016 but since the steps in this method can be difficult to follow for non-specialists I find the diagram helps to easily visualize the method.

Question 13: Line 202: I find Table S1 rather important for this MS. What about involving it to the main MS but not only in supplementary information?
Answer 13: I'm not convinced I think this table is better in extra method.

Question 14: 4. Discussion: Line 301: I think it should be GF17-1A and 1C in the parenthesis.
Answer 14: ok

Question 15: Line 309: (our results) data not shown and presented? If so, please mention or indicate where this information comes from. In the same line, it would be better to mention some of the previous studies showing differences too.

Answer 15: Ok

Question 16: I recommend changing the titles for the section 4.2 and 4.3 to ": : :T1/foraminifera habitat in relation with the nitrate micro-distribution: : :" since there might be other factors having an impact on the ecology of these species, it would be better to keep the focus on nitrate and oxygen in these sections of the discussion.

Answer 16: I suggest to merge the two parts 4.2 and 4.3
4.3 The foraminifera ecology considering the nitrate micro-distribution
Inside first paragraph about oxic station and a second paragraph about hypoxic station.

Question 17: Line 395: once again discussion on benthic N loss contribution to eutrophication: I think this needs a broader discussion and requires some references. Moreover, does N2 flux from sediment promote N2 fixation, and thus, e.g., cyanobacterial activity? Are there studies focusing on N2 fix vs N loss in Gullmar Fjord or similar settings? I think considering these would improve the discussion significantly.

Answer 17: it's difficult to answer this question without getting too speculative
The question here suggests that nitrogen supply via benthic denitrification of the forams could be captured by $N_2$-fixing cyanobacteria and participate in their development. Significant cyanobacteria blooms are already known in the Baltic Sea (Boesch 2003 Swedish agency report). In the Gullmar fjord there are few studies on cyanobacteria (Croot, 2003) the evolution of $N_2$-fixation by these cyanobacteria in Gullmar Fjord is not obvious and lack of data. Benthic denitrification of the forams may participate in the N pool to be fixed by cyanobacteria but I think this hypothesis is too speculative, then cyanobacteria in Gullmar Fjord do not appear to be a major threat to the system at this time.

Constance Choquel
constance.choquel@gmail.com

Dear Referee 2,
Thank you for your constructive comment and your interest in our work. I agree with the majority of the suggestions that you bring to our study. The status of *Nonionella* sp T1 remains unclear. I am to follow the recommendations made by V. Bouchet by introducing *Nonionella* sp. T1 as Non-Indigenous Species then, in discussion I will discuss its invasiveness in Gullmar Fjord. Indeed, the dominance of *Nonionella* sp. T1 could be harmful for the Foraminifera diversity species. I am aware that this study must be followed by a long bio-monitoring > 63 µm (seasonal, different depths stations) to validate the ongoing change in Gullmar Fjord fauna.

Question 1: The title of the study implies, that the work focuses on total nitrate uptake of a specific benthic foraminifer. However, the emphasis of the first part in the discussion of this study implies a thorough taxonomic investigation of the Fjord fauna, which is not the case in this study. I agree with the authors, that there is an ongoing change in the benthic foraminiferal community of the Gullmar Fjord. But to verify
this trend and to discuss its consequences, a longer-term monitoring study observing seasonal fluctuations
of the benthic foraminiferal community together with environmental parameters at several stations within
the fjord is necessary. Further, a more detailed comparison with previous literature would be necessary. I
think the authors should point out, that such monitoring studies (including the 63 – 125µm size fraction)
are important for the future, specifically considering the new observations of this study.
Answer 1: I agree that a long monitoring would be necessary to validate the change in fauna and include
a study with a smaller fraction.

Question 2: I agree with the author of the short comment considering the invasive status of *Nonionella*
sp. T1. Certainly, this species is proven to be non-indigenous. However, the actual invasive status of this
species is not proven yet. It is not yet clear, if the occurrence of Nonionella sp. T1 is responsible for the
disappearance of any other species in the Fjord, nor is there any evidence, that this species is harmful for
the ecosystem of the Gullmar Fjord. On the contrary, the authors point out, that this species might even
be of advantage for the trophic status of the fjord. It is important to stick with correct ecological
terminology to avoid confusion in further research. I would recommend to change the term 'invasive' to
'non-indigenous'.
Answer 2: I agree with V. Bouchet comment. I will introduce *Nonionella* sp. T1 as a Non-Indigenous
Species (Deldick et al., 2019). Then, in the discussion I will mention the invasive character of this species
in the Gullmar Fjord in view of its strong increase in density at the entrance to the Fjord. There is no
evidence that *Nonionella* sp. T1 can harm the ecosystem, however *Nonionella* sp. T1 could affect the
fauna of foraminifera. Indeed, the specific richness (S) and the Shannon index (H) decrease with sediment
depth sediment in the GF17-3 station while the dominance due to Nonionella increases (see graphs GF17-
3A and 3C). In the hypoxic station, the dominance is driven by *Cassidulina laevigata* and *Bulima
marginata* which dominated the fauna.

[Figure]

Additionally, I would like to add a few technical corrections and minor remarks:

Introduction:

Question 3: Line 29: 'and thereby to survive' should be 'and thereby survive'

Answer 3: ok

Question 4: Line 32: 'This study focus on...' should be 'This study focuses on...'

Answer 4: Ok

Material and Methods:

Question 5: Line 127: 'Fixed samples were sieved and the > 100 m fraction was examined...' Did you remove any larger meiofauna e.g. by sieving through a larger sieve (5 mm, 2 mm, 1mm)? If so, this should be mentioned too, since adults of larger denitrifying genera e.g. *Globobulimina* often cannot pass through a 1 mm sieve.

Answer 5: the sieves used are

| >355 | 355-150 | 150-125 | 125-100 |
|------|---------|---------|---------|

No 1 mm sieve was used there should be no loss of *Globobulimina*.

Discussion:

Question 6: Line 292: I would consider to change the title of this section into something like: 'Abundance of Nonionella sp. T1 in comparison with other species'

Answer 6: yes I will change the title to be more careful about the change of fauna.

Question 7: Line 315: I think there is something a little bit wrong with this sentence. Should it be something like: 'That the foraminiferal fauna described in the present study differs, is the consequence...'

Answer 7: I will rewrite better this sentence.

Question 8: Line 327: Did Polodova Asteman and Schönfeld (2015) sample the same location at the oxic part of the fjord?

Answer 8: No, they sampled in the deep Alsback station which was oxic at the time of the sampling in August 2013 and July 2014. They sampled a station in the Skagerrak near the mouth of the fjord in June 2013, I compared my oxic station with this data out of the Fjord.

Question 9: Line 359: Could propagules also be a reason for the survival or re-appearance of the non-denitrifying species in the hypoxic part of the fjord?

Answer 9 : Yes, propagules can disperse and reproduce when environmental conditions are favourable according to Alve and Goldstein, 2003. However, there is no change in density of *Nonionella* sp. T1 at the Alsback station from the densities found by Polovodova Asteman and Schönfeld (2015). It would be interesting to look again at this Alsback station to see if there is an evolution of the densities of *Nonionella* sp. T1 and if there is a seasonality of denitrifying foraminifera depending on the oxygenation conditions (hypoxic vs oxic).

Question 10: Line 392: I would be careful with this consideration, because other well oxygenated areas of the Fjord might be dominated by other species - depending on depth or other environmental parameters.

Answer 10: Yes to bring more weight to this hypothesis it would be necessary to make several oxic stations at different depths in the Fjord.

Question 11: Figure 6: It should be 'Depth (mm)' for GF17-3A and 3C and GF17-1A and 1C and not Depth (cm).
Answer 11: ok

---

## Author Response (AR2)

constance.choquel@gmail.com on behalf of the coauthors

Dear Editor and referees,

Please find the new version of the paper "Denitrification by benthic foraminifera and their contribution to N-loss from a fjord environment". The main corrections performed to the manuscript are highlighted in pink.

Briefly, the corrections requested related to improving the quality of the English manuscript. This manuscript was proofread by 3 English-speaking coauthors.

- The Abstract of the manuscript has been clarified and improved.

- The Introduction and the Material and Methods have been lightened and English improved, while keeping the content remains unchanged with all comments from reviewers taken into account.

- The internet address of SMHI has been included in the bibliographic list.

- Few corrections were made on the Results only to clarify the reading.

- A clarification was made on Figure 5 and in the text. Indeed, the first denitrification zone of the GF17-3 station is 1.2 to 3.5 cm deep.

- Four bibliographic references suggested by the coauthors have been added.

For more details on minor changes please refer to the manuscript with corrections.

Best regards.
Constance Choquel

[revised manuscript text omitted]